# FALQON: Accelerating LoRA Fine-tuning with Low-Bit Floating-Point Arithmetic

**Kanghyun Choi**     **Hyeyoon Lee**     **SunJong Park**     **Dain Kwon**     **Jinho Lee**

Department of Electrical and Computer Engineering
Seoul National University
{kanghyun.choi, hylee817, ryan0507, dain.kwon, leejinho}@snu.ac.kr

## Abstract

Low-bit floating-point (FP) formats, such as FP8, provide significant acceleration and memory savings in model training thanks to native hardware support on modern GPUs and NPUs. However, we analyze that FP8 quantization offers speedup primarily for large-dimensional matrix multiplications, while inherent quantization overheads diminish speedup when applied to low-rank adaptation (LoRA), which uses small-dimensional matrices for efficient fine-tuning of large language models (LLMs). To address this limitation, we propose FALQON, a novel framework that eliminates the quantization overhead from separate LoRA computational paths by directly merging LoRA adapters into an FP8-quantized backbone *during fine-tuning*. Furthermore, we reformulate the forward and backward computations for merged adapters to significantly reduce quantization overhead, and introduce a row-wise proxy update mechanism that efficiently integrates substantial updates into the quantized backbone. Experimental evaluations demonstrate that FALQON achieves approximately a $3\times$ training speedup over existing quantized LoRA methods with a similar level of accuracy, providing a practical solution for efficient large-scale model fine-tuning. Moreover, FALQON's end-to-end FP8 workflow removes the need for post-training quantization, facilitating efficient deployment. Code is available at `https://github.com/iamkanghyunchoi/falqon`.

## 1  Introduction

Recent advancements in large language models (LLMs) have demonstrated remarkable progress across a wide range of natural language understanding and generation tasks. Despite these achievements, the massive parameter counts in LLMs require prohibitively large computational and memory resources, posing significant challenges for both training and deployment. In many real-world settings, even those with a reasonable amount of computational resources, training or fine-tuning such massive models becomes especially challenging [6, 13, 40].

One promising way to reduce the computational burden of LLM fine-tuning is through low-precision floating-point (FP) [29, 33, 34], such as FP8, which can theoretically double the throughput of FP16 matrix multiplication thanks to the hardware support of modern GPUs and NPUs. However, it is known that FP8 quantization primarily accelerates large-dimensional matrix multiplication, while small-dimensional computations suffer from unexpected slowdown. This is due to quantization overhead, which requires a reduction operation followed by element-wise scaling of a given tensor. Therefore, FP8 quantization becomes effective when computational gains surpass the overhead, which empirically occurs when each dimension of the matrices exceeds approximately 4K elements [43].

39th Conference on Neural Information Processing Systems (NeurIPS 2025).

These quantization overheads are especially critical when it comes to fine-tuning with low-rank adaptation (LoRA) [20]. LoRA inserts small-dimensional trainable low-rank matrices (adapters) to capture task-specific knowledge, significantly reducing the memory cost by using fewer trainable parameters. However, for matrices with small dimensions, such as LoRA adapters, the overhead incurred by FP8 quantization can outweigh the benefits from FP8 multiplications. Also, separate forward and backward paths for LoRA introduce a larger number of quantization operations, worsening the overall overhead. In our preliminary analyses (Section 4), we show that applying FP8 quantization to LoRA introduces significant quantization overhead, limiting speedup. This slowdown in LoRA poses critical challenges in practical scenarios, where numerous adapters must be trained to support personalization [59], multi-task learning [28], and rapid updates in dynamic, user-specific environments (see Section 3.1). Thus, efficient acceleration of LoRA fine-tuning is essential to enable timely, scalable, and cost-effective deployment of LLMs under practical computational constraints.

To address this, we propose FALQON (**FP8-A**ccelerated **L**oRA **Q**uantizati**on**), a novel framework designed specifically to accelerate FP8-based quantized LoRA fine-tuning by reducing quantization overheads. Instead of separate LoRA adapters, FALQON merges adapters directly into the FP8 backbone *during fine-tuning*, leveraging the initial quantization error as an implicit LoRA initialization (*melded LoRA*) to eliminate extra quantization steps. Additionally, we reformulate both forward and backward computational paths for efficient gradient calculation of the merged adapters. A row-wise proxy update mechanism selectively applies substantial weight updates to the backbone, avoiding ineffective updates that vanish under low-bit quantization and further enhancing overall efficiency.

Through extensive experiments on various tasks, we demonstrate that FALQON achieves up to $3\times$ faster fine-tuning compared to quantized LoRA baselines, while maintaining comparable accuracy. Moreover, the end-to-end FP8 workflow of FALQON eliminates the need for post-training quantization, facilitating efficient deployment. Our key contributions are summarized as follows:

- We analyze FP8 quantization overhead and show that existing FP8 quantization methods primarily target large-dimensional matrix multiplications, resulting in substantial overhead and limited speedups when directly applied to LoRA's small-dimensional adapters.

- We propose FALQON, a novel framework that merges LoRA adapters into an FP8-quantized backbone during fine-tuning, significantly reducing quantization overhead.

- We reformulate forward and backward paths for efficient gradient computation of merged adapters and introduce a row-wise proxy update mechanism that selectively integrates substantial updates, avoiding unnecessary weight modifications under low-bit quantization.

- We empirically demonstrate that FALQON achieves up to $3\times$ faster fine-tuning compared to existing methods, providing detailed breakdown analyses to identify sources of acceleration, while maintaining comparable accuracy across comprehensive evaluations.

## 2   Background

### 2.1   Low-rank adaptation

Low-rank adaptation (LoRA) aims to reduce trainable parameters during fine-tuning by decomposing a parameter update into low-rank matrices. Specifically, let $W \in \mathbb{R}^{m \times n}$ be the weight matrix and $x \in \mathbb{R}^{n \times d}$ be the activation. LoRA represents the weight update $\Delta W$ via a product of low-rank matrices $B \in \mathbb{R}^{m \times r}$ and $A \in \mathbb{R}^{r \times n}$, where $r \ll \min(m, n)$. Then, the forward pass becomes:

$$(W + \Delta W)x = (W + BA)x = Wx + BAx. \tag{1}$$

The low-rank product $BAx$ serves as an efficient approximation of $\Delta Wx$, reducing trainable parameters to $mr + nr$ (compared to $mn$), which is essential for fine-tuning large models under resource constraints. During backpropagation, the gradients of the loss $\mathcal{L}$ with respect to A and B are:

$$\frac{\partial \mathcal{L}}{\partial A} = B^{\top} \frac{\partial \mathcal{L}}{\partial O} x^{\top}, \quad \frac{\partial \mathcal{L}}{\partial B} = \frac{\partial \mathcal{L}}{\partial O} x^{\top} A^{\top}, \tag{2}$$

where $O = (W + BA)x$ is the layer's output. By keeping $r \ll \min(m, n)$, LoRA enables efficient fine-tuning of large models without requiring full parameter updates.

Quantized LoRA approaches [12, 26, 35, 52] further reduce memory usage by applying weight-only quantization to backbone weights. As the weight parameters of LLMs often span tens to hundreds of

gigabytes (*e.g.*, approximately 14GB for 7B parameters and 140GB for 70B parameters in FP16), these methods quantize the backbone weights to low-precision formats, as follows:

$$Wx + BAx \approx DQ(Q(W))x + BAx, \tag{3}$$

$$Q(W) = \widetilde{W}, \quad DQ(\widetilde{W}) \approx W, \tag{4}$$

where $Q(\cdot)$ quantizes weights to low-precision representations ($W_Q$), and $DQ(\cdot)$ reconstructs the high-precision approximation of quantized weight $\widetilde{W}$. These additional quantization and dequantization operations inherently introduce computational overhead, slowing down the fine-tuning. Our proposed approach is designed to mitigate quantization overhead and accelerate LoRA fine-tuning through low-precision arithmetic. By quantizing both weights and activations with minimal overhead, FALQON enables efficient matrix multiplication in contrast to weight-only quantization.

## 2.2 Low-precision floating-point quantization

Modern hardware accelerators (*e.g.*, NVIDIA Hopper [32] and Blackwell [31], AMD CDNA 3 [1], Google Ironwood TPU [15]) increasingly support low-precision FP arithmetic, significantly boosting efficiency in training and inference. In general, FP datatypes are often denoted by an $E\alpha M\beta$ format, which consists of a single sign bit, $\alpha$ exponent bits, and $\beta$ mantissa (fraction) bits.

Low-precision FP8 formats (e.g., E4M3 or E5M2) have a narrow representable range. Therefore, a higher-precision tensor $X$ (e.g., FP32 or FP16) must be scaled by a per-tensor scaling factor $s_X$. The scaled tensor $s_X \cdot X$ is then rounded to FP8, as follows:

$$\widetilde{X} = Q_{fp8}(X) = \lfloor X \cdot s_X \rceil, \quad s_X = \frac{\max(|X|)}{\max(\text{FP8 dtype})}, \tag{5}$$

where $\widetilde{X}$ is the quantized tensor, $Q_{fp8}(\cdot)$ is the 8-bit floating-point quantization function, $\lfloor \cdot \rceil$ is round-to-nearest operation, and $\max(\text{FP8 dtype})$ is the maximum representable value for the chosen datatype, *i.e.*, 448 for E4M3 and 57344 for E5M2. The low-precision multiplication can be performed natively on hardware optimized for FP8, after which the result is rescaled by $1/(s_W s_x)$:

$$Wx \approx \overbrace{\widetilde{W}\widetilde{x}}^{\text{FP8 matmul}} / \underbrace{s_W s_x}_{\text{Rescale}}, \tag{6}$$

$$\because W \approx DQ_{fp8}(\widetilde{W}) = \widetilde{W}/s_W, x \approx DQ_{fp8}(x) = \widetilde{x}/s_x. \tag{7}$$

Although FP8 computations can be significantly faster, the overhead of computing the per-tensor scale $s_X$ (which involves a reduction operation $\max(|X|)$) and element-wise scale operation (Equation (5)) may overshadow the speedup. The FP8 quantization overhead (amax reduction and element-wise scaling in Equation (5)) scales quadratically ($O(n^2)$) with matrix size $n$. This overhead is memory-bound, requiring repeated reads and writes for each tensor element, dominating execution time for smaller matrices. Therefore, FP8 arithmetic only becomes beneficial at larger matrix sizes (empirically around $n \geq 4096$), where the cubic complexity ($O(n^3)$) of matrix multiplication outweighs the quantization overhead. Detailed analysis of these trade-offs is provided in Section 4.

## 3 Related Work

### 3.1 Low-rank adaptation

LoRA [20] significantly reduces resource usage by updating only low-rank adapters during fine-tuning [7, 8, 39, 46] or even training from scratch [27]. Recently, there has been an increased interest in scenarios involving the training of numerous LoRA adapters. **Personalization**: RecLoRA [59] and PLoRA [55] train an independent LoRA for different users for recommendation. Also, Liu *et al.* [28] trains multiple LoRA for personalized RLHF training. **Multitask Learning**: Mixture of LoRA Experts [49] trains multiple task-specific LoRA adapters and arranges these adapters. **Domain Adaptation**: LoRA adapters trained with multiple domain are used to handle various contexts [11]. **Multilingual Summarization**: Whitehouse *et al.* [47] demonstrate the application of LoRA for multilingual summarization. Given these diverse and practical applications requiring numerous adapters, accelerating LoRA fine-tuning holds significant practical value.

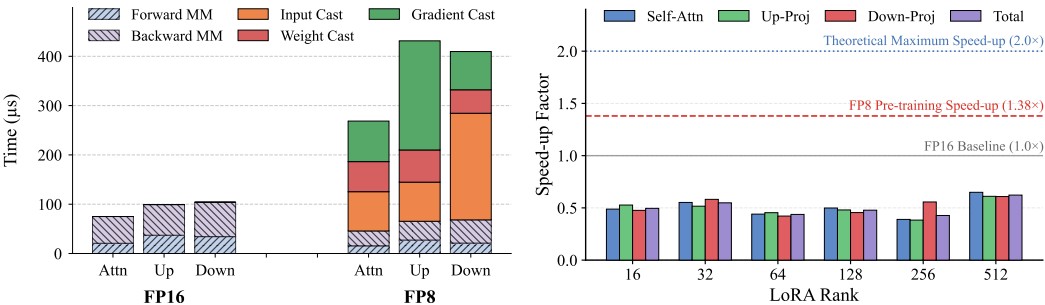

(a) Breakdown analysis of LoRA computation.  (b) Speedup comparison of various LoRA ranks

Figure 1: Preliminary experiments on quantization overhead for LoRA ($r = 64$) using linear layers from LLaMA-7B (self-attention, up-projection, down-projection). (a) FP8 reduces computation time. However, the quantization overhead dominates the computation and results in nearly fourfold higher latency. (b) FP8 shows consistently lower throughput than FP16 across ranks. These results demonstrate that FP8 quantization overhead cancels the expected acceleration in LoRA fine-tuning.

Several quantized LoRA approaches have recently been proposed for memory efficiency of LoRA fine-tuning. QLoRA [12] uses weight-only NF4 quantization to significantly reduce memory usage, while QA-LoRA [52] leverages group-aware INT4 quantization for efficient deployment. The following approaches improve quantization by employing integer linear programming [16], SVD-based adapter initialization [26], dynamic rank allocation [21, 25], integer-based LoRA quantization [17], and information recovery strategies [35]. However, these methods primarily focus on memory efficiency while overlooking fine-tuning speed. Consequently, they often exhibit slower training than standard LoRA due to substantial quantization overhead. Moreover, since these methods rely on weight-only quantization, they do not leverage hardware acceleration from quantization. In contrast, our proposed method successfully reduces memory consumption while significantly accelerating LoRA fine-tuning, demonstrating superior efficiency compared to previous quantized LoRA approaches.

## 3.2 Quantized training with FP datatypes

Early quantized training methods primarily utilize low-bit integers [2, 48, 50, 56–58]. Recently, FP quantization has gained attention due to native hardware support on GPUs and NPUs, providing wider dynamic range and improved precision than integers [45]. Recognizing different numerical ranges of gradients and activations, Sun *et al.* [41] proposed hybrid FP8 training, using E5M2 for gradients and E4M3 for activations and weights. Other studies have addressed the limited dynamic range of FP8 by adjusting exponents [24] or modifying special-value representations [29].

Following the release of NVIDIA's Hopper architecture with native FP8 support [33], recent studies have actively pursued practical acceleration of FP8-based training. FP8-LM [34] introduced mixed-precision training, keeping only master weights and second-order momentum at higher precision. Fishman *et al.* [14] further explored large-scale FP8 training stability and outliers, proposing modified SwiGLU. Meanwhile, unit-scaling methods [4, 5] simplified FP8 training by determining optimal scaling at initialization. Despite these advances, they target large-dimensional matrices of LLM training. In contrast, our method specifically accelerates the smaller-dimensional matrix multiplications in LoRA fine-tuning, achieving significant acceleration for the LoRA adapters.

## 4 Preliminary analysis

LoRA fine-tuning commonly involves small-dimensional matrices. In such cases, the computational overhead of quantization outweighs the speedup. This is because the separate LoRA path requires additional quantization for small matrices. As illustrated in Figure 2, the LoRA fine-tuning introduces three additional quantization processes for two small-tensor multiplications per iteration compared to the direct training of the backbone. From the FP8-quantized $\widetilde{X}$, a LoRA fine-tuning requires quantizations of $\widetilde{A}$ and $\widetilde{B}$ from their newly updated weights, and another quantization for $O_A$, the intermediate activation. Because $\widetilde{A}$ and $\widetilde{B}$ are both small, their speedup from using FP8 arithmetic cannot offset the quantization overheads.

In Figure 1a, we investigate the impact of quantization overhead on small-dimensional LoRA ($r$=64) by breaking down the computation time of FP8 and FP16 arithmetic using an RTX 4090 GPU. Although FP8 arithmetic significantly reduces the computation time, the quantization overhead substantially outweighs these benefits, exhibiting approximately three to four times higher latency. This issue persists even when employing adapters with larger ranks. In Figure 1b, we compare the computation time of FP8 and FP16 LoRA adapters across various ranks. For the reference, we show empirical speedup of pretraining of LLaMA-7B model with TorchAO [43]. The results indicate a consistent slowdown of FP8 compared to FP16, showing roughly half the throughput across typical LoRA ranks (16–128). Even at significantly higher ranks (*e.g.*, 256 or 512), FP8 adapters continue exhibiting notable degradation. Please refer to Section C in the Appendix for the results from the larger ranks. These preliminary results indicate that the inherent quantization overhead in FP8 arithmetic outweighs its computational advantages, emphasizing the need to significantly reduce quantization overhead to fully leverage FP8 arithmetic to accelerate LoRA fine-tuning.

# 5 Proposed methods

In this section, we introduce FALQON, a novel framework for LoRA fine-tuning using FP8 arithmetic with significant speedups that were previously unavailable due to the quantization overheads. Motivated by inefficiencies arising from separate LoRA computations, our core idea is to directly *meld* LoRA adapters into the quantized backbone weights. This eliminates separate small-tensor LoRA operations in the forward pass. Because we let the weight updates still be done only through the LoRA, even though it is melded into the backbone, we maintain the memory benefits of LoRA fine-tuning. In the remainder of this section, we describe how we initialize and quantize the melded LoRA adapters (Section 5.1), how to compute gradients without explicitly storing the adapter weights (Section 5.2), and propagate the gradient of the adapters to the quantized melded backbone (Section 5.3).

## 5.1 Melded LoRA: Merging backbone and LoRA from the start

We propose *melded LoRA*, a method that merges LoRA adapters directly into the quantized backbone weights to eliminate separate LoRA adapter paths. Motivated by Section 4, melded LoRA merges the LoRA adapter into the backbone weights during fine-tuning, removing the separate paths.

The key idea of melded LoRA is interpreting quantization error $\Delta_Q W$ as an additive tensor to the original high-precision weight $W$. Quantized backbone weights $\widetilde{W}$ inherently have quantization error $\Delta_Q W$ compared to the original high-precision backbone weights $W$, as follows:

$$DQ_{fp8}(\widetilde{W}) + \Delta_Q W = W, \qquad (8)$$

where $DQ_{fp8}(\cdot)$ denotes the FP8 dequantization function.

Then, we reformulate Equation (8) as follows:

---

**Algorithm 1** FALQON: Initialization

1: **Input:** rank $r$, pretrained backbone $W$, quantization function $Q(\cdot)$
2: **for** $l = 1, \ldots, L$ **do**
3: $\quad \widetilde{W}_l, s_{W_l} \leftarrow Q(W_l)$
4: $\quad \Delta W_l \leftarrow W_l - \widetilde{W}_l/s_{W_l}$
5: $\quad \widehat{B}_l, \widehat{A}_l \leftarrow SVD(\Delta W_l, r)$
6: $\quad \widetilde{A}_l \leftarrow \lfloor A_l \cdot s_{W_l} \rceil$
7: $\quad \widetilde{W}' \leftarrow [\widetilde{W}^\top \mid \widetilde{A}^\top]^\top$
8: $\quad \Delta B\text{uffer}_l \leftarrow \{0\}^{C_{out} \times r}$
9: **end for**

---

$$DQ_{fp8}(\widetilde{W}) = W - \Delta_Q W \approx W + SVD(-\Delta_Q W; r) = W + \widehat{B}\widehat{A}, \qquad (9)$$

where $\widehat{A}, \widehat{B}$ are low-rank matrices from the rank-$r$ SVD of $-\Delta_Q W$. This reformulation interprets the quantized backbone $DQ_{fp8}(\widetilde{W})$ as implicitly including the LoRA adapter $\widehat{B}\widehat{A}$, allowing a single computation of $DQ_{fp8}(\widetilde{W})x$ to produce the LoRA-adapted output directly (*i.e.*, $Wx + \widehat{B}\widehat{A}x$), thus allowing efficient forward passes without additional adapter computations. Note that although existing works [16, 26] similarly leverage quantization error, they require a separate LoRA to restore the original weights $W$, thus they do not address computational overhead. In contrast, our method effectively removes overhead, significantly enhancing efficiency (Figure 2).

## 5.2 Efficient gradient computation for Melded LoRA

Unlike conventional LoRA, melded LoRA integrates adapters implicitly into backbone weights. Thus, adapter gradients are not explicitly computed by standard autograd operations. We address this by

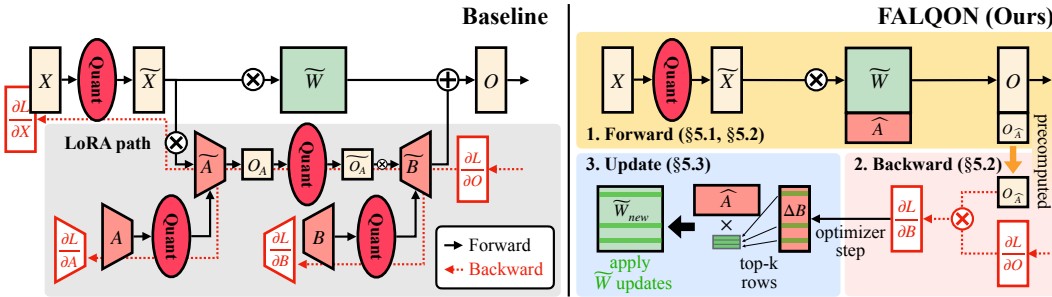

Figure 2: Comparison between conventional LoRA fine-tuning and the proposed FALQON framework. **Left:** LoRA fine-tuning introduces redundant quantization for small-dimensional matrices ($\widetilde{A}$, $\widetilde{B}$, and $O_A$), where the quantization overhead outweighs the computational benefit of FP8 arithmetic. **Right:** FALQON eliminates these overheads by directly *melding* LoRA into the quantized backbone, allowing the forward pass to be executed without separate small-tensor operations.

proposing an effective gradient computation method, specifically designed for the merged structure of melded LoRA, for further reducing overhead.

To naively compute LoRA gradients, one could reconstruct low-rank matrices $\widehat{A}$ and $\widehat{B}$ each iteration from the quantization error $\Delta_Q W$ using SVD:

$$\widehat{A}, \widehat{B} = SVD(W - DQ_{fp8}(\widetilde{W}); r). \qquad (10)$$

With these approximations, we compute gradients following Equation (2).

However, conventional gradient computations of LoRA involve small-dimensional matrix multiplications, which incur significant quantization overhead. To mitigate this, we reformulate the gradient calculation of $B$ in Equation (2), as follows:

$$\frac{\partial \mathcal{L}}{\partial B} = \frac{\partial \mathcal{L}}{\partial O} x^\top A^\top = \frac{\partial \mathcal{L}}{\partial O} (Ax)^\top. \qquad (11)$$

**Algorithm 2** FALQON: Gradient Computation

---

1: **Input:** Training data $\{d_i\}_{i=1}^N$, rank $r$, Quantized backbone $\widetilde{W}$, quantization function $Q_{fp8}(\cdot)$
2: **for** $i = 1, ..., N$ **do**
3:    $x_0 \leftarrow d_i$
4:    **for** $l = 1, ..., L$ **do**
5:       $\widetilde{x}_{l-1}, s_{x_{l-1}} \leftarrow Q_{fp8}(x_{l-1})$
6:       $O_{merged} \leftarrow \widetilde{W}'_l \widetilde{x}_{l-1} / s_{W_l} s_{x_{l-1}}$
7:       $O, O_{A_l} \leftarrow O_{merged}$
8:       $x_l \leftarrow O$
9:       $ctx.save\_for\_backward(O_{A_l})$
10:    **end for**
11:    Backward and Update (Algorithm 3)
12: **end for**

---

Then, as the computation of $Wx$ and $Ax$ shares the input $x$, we integrate the computation of $Ax$ into the quantized forward pass by concatenating $\widehat{A}$ to the quantized backbone weights $\widetilde{W}$:

$$\widetilde{W}' = \begin{bmatrix} \widetilde{W} \\ \widetilde{A} \end{bmatrix} \in \mathbb{R}^{(m+r) \times n}, \quad \widetilde{A} = Q_{fp8}(\widehat{A}; s_W) = \lfloor \widehat{A} \cdot s_W \rceil. \qquad (12)$$

This enables simultaneous computation of both backbone output $O$ and intermediate output $O_{\widehat{A}} = \widehat{A}x$ for gradient computation (Equation (11)) in a single forward without additional quantization:

$$O_{merged} = \widetilde{W}' \widetilde{x} / s_W s_x = \begin{bmatrix} O \\ O_{\widehat{A}} \end{bmatrix} \in \mathbb{R}^{(m+r) \times d}, \qquad (13)$$

where $O \in \mathbb{R}^{m \times d}, O_{\widehat{A}} \in \mathbb{R}^{r \times d}$. By embedding and freezing $\widehat{A}$ in the quantized backbone, we eliminate any additional gradient computations for the LoRA path. We only compute gradients of $\widehat{B}$, which is empirically sufficient for effective fine-tuning [54]. Although concatenating $\widehat{A}$ to the forward pass slightly increases the computation, the additional cost is negligible due to its small dimension ($r$). See lines 3–7 of Algorithm 1 for initialization and Algorithm 2 for computations.

## 5.3 Row-wise updates of quantized weights using proxy buffer

Given the merged structure of melded LoRA, gradients computed for the approximated $\widehat{B}$ must be directly applied to the merged backbone weights, because of the absence of a separate LoRA path. Formally, the weight updates from $B$ matrix are:

$$W + (B + \Delta B)A = (W + BA) + \Delta BA = \widetilde{W} + \Delta BA. \qquad (14)$$

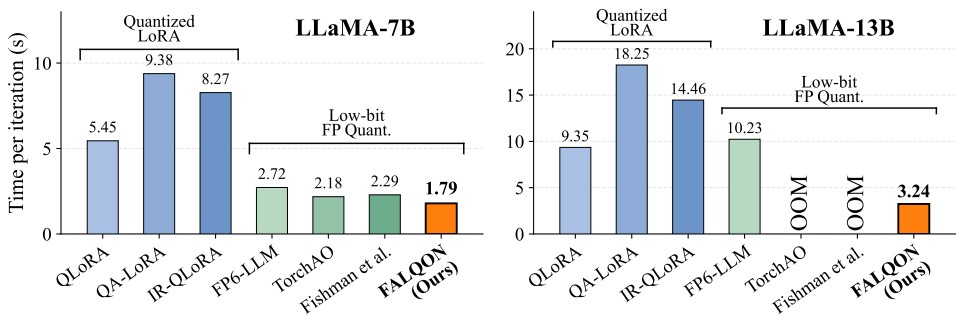

Figure 3: Overall computational cost comparison of baselines and FALQON (lower is better).

This indicates that the merged backbone weights $\widetilde{W}$ can be updated only from updates $\Delta B$ and $A$, without the need to store full-precision $W$ and $B$. From this, we introduce a proxy update matrix $\Delta B$uffer, which stores only changes of the $B$ matrix. Specifically, instead of updating $\widehat{B}$, we store gradient-driven changes in the proxy matrix $\Delta B$uffer by recording changes of $B$ from the optimizer step. Then, by storing $\Delta B$, we can apply the updates of melded LoRA into merged backbone.

---

**Algorithm 3** FALQON: Backward and Update

1: **Input:** learning rate $\eta$, Quantized backbone $\widetilde{W}$, Optimizer, gradient $\partial\mathcal{L}/\partial x_L$
2: **for** $l = L, ..., 1$ **do**
3: $\quad \frac{\partial\mathcal{L}}{\partial B_l} \leftarrow \frac{\partial\hat{\mathcal{L}}}{\partial x_l} O_{A_l}^T$ ▷ Equation (11)
4: $\quad \Delta B$uffer$_l \leftarrow Optimizer(\frac{\partial\mathcal{L}}{\partial B_l}, \eta)$
5: $\quad \mathbf{k} \leftarrow topk(\sum_j |\Delta B_{i,j}|; k),$
6: $\quad \widetilde{W}[\mathbf{k}] = \widetilde{W}[\mathbf{k}] + \Delta B$uffer$[\mathbf{k}]A$
7: **end for**

---

To efficiently incorporate impactful updates of $\Delta B$uffer into the backbone weights, we selectively propagate large updates. Given that $\widetilde{W}$ is represented in low-bit, minor changes often result in no effective update. Therefore, we identify the indices $\mathbf{k}$ with the top-$k$ largest updates in $\Delta B$uffer and selectively apply only significant changes:

$$\mathbf{k} = topk(\sum_j |\Delta B_{i,j}|; k), \quad \widetilde{W}[\mathbf{k}] = \widetilde{W}[\mathbf{k}] + \Delta B\text{uffer}[\mathbf{k}]A. \qquad (15)$$

Since $k \ll m$ (output channels), this selective update method (Algorithm 3) efficiently integrates substantial updates into $\widetilde{W}$, while minimizing redundant computations and improving overall efficiency.

## 6 Evaluation

### 6.1 Experimental settings

We fine-tune LLaMA-7B and 13B on the Alpaca [42] and OASST1 [23] datasets, then evaluate on the Massively Multitask Language Understanding (MMLU [19]) benchmark, which measures knowledge and reasoning across a diverse set of domains. In addition, we assess our models on HellaSwag [53], PIQA [3], WinoGrande [36], ARC [10], BoolQ [9], and OpenBookQA [30] for commonsense QA, following QA-LoRA's five-shot evaluation protocol via the `lm-eval-harness` framework. Unless otherwise noted, all experiments are conducted on a single 24GB RTX 4090 GPU. We compare our method (FALQON) against quantized LoRA baselines (QLoRA [12], QA-LoRA [52], IR-QLoRA [35]) and FP quantization methods (FP6-LLM [51], Fishman *et al.* [14] and TorchAO [43]). We follow the settings of QLoRA: a Paged AdamW optimizer with a batch size of 16, learning rate of $2e$-5, and 1,875 training steps. Refer to the Appendix for the detailed settings.

### 6.2 Comparison of fine-tuning efficiency and scalability

We evaluate the fine-tuning efficiency of FALQON against quantized LoRA and low-bit FP baselines (Figure 3). Quantized LoRA methods primarily reduce memory usage but exhibit higher computational overhead, enabling larger (13B) models to fit into GPU memory but at significantly slower speeds. Low-bit FP methods achieve faster computation due to accelerated matrix multiplication. However, methods like TorchAO and Fishman *et al.* [14] maintain high-precision tensors, thus failing to fine-tune the 13B model due to out-of-memory (OOM) error. In contrast, FALQON simultaneously accelerates FP8 computation and reduces memory cost, resulting in the lowest iteration time for both 7B and 13B models, achieving superior efficiency. See Section G for comparisons on other GPUs.

Table 1: Comparison of time per step (lower is better) and MMLU (higher is better)

(a) LLaMA-7B and 13B on Alpaca dataset

| | Metric | QLoRA | QA-LoRA | IR-QLoRA | FALQON (Ours) |
|---|---|---|---|---|---|
| | Time / Step (s) | 5.45 | 9.44 | 8.27 | **1.80 (3.02×)** |
| | #T. Params. | 160M | 89M | 89M | 80M |
| 7B | Humanities | 0.3095 | 0.3413 | 0.3224 | 0.3322 |
| | STEM | 0.2902 | 0.3137 | 0.2997 | 0.3086 |
| | Social | 0.3507 | 0.3711 | 0.3659 | 0.3858 |
| | Other | 0.3685 | 0.4007 | 0.3762 | 0.3795 |
| | Average | 0.3272 | 0.3548 | 0.3388 | 0.3491 |
| | Time / Step (s) | 9.37 | 18.02 | 14.46 | **3.26 (2.87×)** |
| | #T. Params. | 250M | 140M | 140M | 125M |
| 13B | Humanities | 0.4253 | 0.4431 | 0.4157 | 0.4408 |
| | STEM | 0.3438 | 0.3834 | 0.3356 | 0.3638 |
| | Social | 0.5096 | 0.5398 | 0.4911 | 0.5414 |
| | Other | 0.5105 | 0.5426 | 0.5092 | 0.5259 |
| | Average | 0.4443 | 0.4729 | 0.4349 | 0.4644 |

(b) LLaMA-7B and 13B on OASST1 dataset

| | Metric | QLoRA | QA-LoRA | IR-QLoRA | FALQON (Ours) |
|---|---|---|---|---|---|
| | Time / Step (s) | 5.45 | 9.38 | 8.34 | **1.79 (3.04×)** |
| | #T. Params. | 160M | 89M | 89M | 80M |
| 7B | Humanities | 0.3367 | 0.3439 | 0.3362 | 0.3373 |
| | STEM | 0.3143 | 0.3159 | 0.3232 | 0.3130 |
| | Social | 0.3884 | 0.3890 | 0.3949 | 0.3776 |
| | Other | 0.3972 | 0.4046 | 0.4010 | 0.3708 |
| | Average | 0.3564 | 0.3609 | 0.3605 | 0.3481 |
| | Time / Step (s) | 9.35 | 18.25 | 15.22 | **3.24 (2.89×)** |
| | #T. Params. | 250M | 140M | 140M | 125M |
| 13B | Humanities | 0.4355 | 0.4387 | 0.4321 | 0.4436 |
| | STEM | 0.3717 | 0.3920 | 0.3765 | 0.3638 |
| | Social | 0.5226 | 0.5499 | 0.5193 | 0.5349 |
| | Other | 0.5272 | 0.5488 | 0.5375 | 0.5288 |
| | Average | 0.4605 | 0.4769 | 0.4620 | 0.4645 |

Table 2: Comparison of low-precision FP quantization methods on Alpaca and OASST1 dataset

| Method | Type | Time / Step (s) | # Trainable Params | Alpaca (MMLU) Hum. | STEM | Social | Other | Avg. | OASST1 (MMLU) Hum. | STEM | Social | Other | Avg. |
|---|---|---|---|---|---|---|---|---|---|---|---|---|---|
| LoRA | FP16 | 2.87 | 160M | 0.3295 | 0.3031 | 0.3717 | 0.3873 | 0.3456 | 0.3401 | 0.3258 | 0.4006 | 0.4102 | 0.3656 |
| TorchAO | FP8 | 2.18 | 160M | 0.3231 | 0.2969 | 0.3679 | 0.3785 | 0.3393 | 0.3273 | 0.3092 | 0.3672 | 0.3869 | 0.3452 |
| FP6-LLM | E2M3 | 2.72 | 160M | 0.2421 | 0.2125 | 0.2171 | 0.2398 | 0.2308 | 0.2448 | 0.2125 | 0.2177 | 0.2411 | 0.2308 |
| FP6-LLM | E3M2 | 2.72 | 160M | 0.2487 | 0.2693 | 0.2532 | 0.2333 | 0.2509 | 0.2423 | 0.2249 | 0.2190 | 0.2411 | 0.2330 |
| Fishman *et al.* | FP8 | 2.29 | 160M | 0.3337 | 0.3108 | 0.3893 | 0.3923 | 0.3537 | 0.3241 | 0.2969 | 0.3773 | 0.3714 | 0.3401 |
| FALQON (Ours) | FP8 | **1.79** | 80M | 0.3322 | 0.3086 | 0.3858 | 0.3795 | 0.3491 | 0.3373 | 0.3130 | 0.3776 | 0.3708 | 0.3481 |

## 6.3 Comparison with quantized LoRA frameworks

We compare the fine-tuning speed and accuracy of our method (FALQON) against quantized LoRA baselines on the Alpaca (Table 1a) and OASST1 (Table 1b) datasets. For LLaMA-7B, FALQON achieves roughly three-fold speedup over QA-LoRA (e.g., 1.80s vs. 9.44s per step on Alpaca), while maintaining competitive MMLU scores. Similarly, when scaling to LLaMA-13B, FALQON significantly outperforms baselines in computational efficiency (3.26s vs. 18.02s per step for QA-LoRA on Alpaca), without meaningful degradation in accuracy. Furthermore, FALQON consistently demonstrates stable performance, avoiding accuracy fluctuations across various categories observed in baseline methods. These results highlight the robustness and practicality of FALQON as an effective and efficient solution for quantized LoRA fine-tuning.

## 6.4 Comparison with FP quantization methods

Table 2 compares the MMLU scores and training speeds of Baseline FP16 LoRA and low-bit FP quantization methods (TorchAO, FP6-LLM, Fishman *et al.* [14] and FALQON). We quantize all linear layers for FP8 methods and quantize the backbone for FP6-LLM similar to QLoRA. FALQON achieves the fastest runtime (1.79s/step) while showing comparable MMLU scores. In particular, FALQON average MMLU score on Alpaca (0.3491) surpasses those of TorchAO (0.3393) and FP6-LLM, and a similar trend is observed on OASST1, achieving the highest accuracy (0.3481). These results underscore the efficiency and effectiveness of FALQON.

Table 3: Comparison on common sense reasoning benchmarks

| Method | ARC-C | ARC-E | BoolQ | HellaSwag | OBQA | PIQA | Winogrande | Average |
|---|---|---|---|---|---|---|---|---|
| LoRA (FP16) | 0.4706 | 0.7844 | 0.8025 | 0.5849 | 0.3684 | 0.7984 | 0.7154 | 0.6464 |
| QLoRA (NF4) | 0.4735 | 0.7891 | 0.7856 | 0.5787 | 0.3660 | 0.7976 | 0.7159 | 0.6438 |
| QA-LoRA (INT4) | 0.4735 | 0.7723 | 0.7511 | 0.5618 | 0.3620 | 0.7867 | 0.7230 | 0.6329 |
| IR-QLoRA (NF4) | 0.4812 | 0.7786 | 0.7902 | 0.5819 | 0.3680 | 0.8009 | 0.7230 | 0.6463 |
| FP6-LLM (*E2M3*) | 0.2125 | 0.2681 | 0.3783 | 0.2616 | 0.1400 | 0.5229 | 0.5075 | 0.3273 |
| FP6-LLM (*E3M2*) | 0.2073 | 0.2647 | 0.3783 | 0.2600 | 0.1020 | 0.5239 | 0.4957 | 0.3188 |
| TorchAO (FP8) | 0.4753 | 0.7870 | 0.7933 | 0.5824 | 0.3640 | 0.7960 | 0.7206 | 0.6455 |
| Fishman *et al.* (FP8) | 0.4804 | 0.7896 | 0.7792 | 0.5826 | 0.3640 | 0.7949 | 0.7222 | 0.6447 |
| FALQON (Ours) | 0.4608 | 0.7786 | 0.7676 | 0.5711 | 0.3420 | 0.7905 | 0.7135 | 0.6320 |

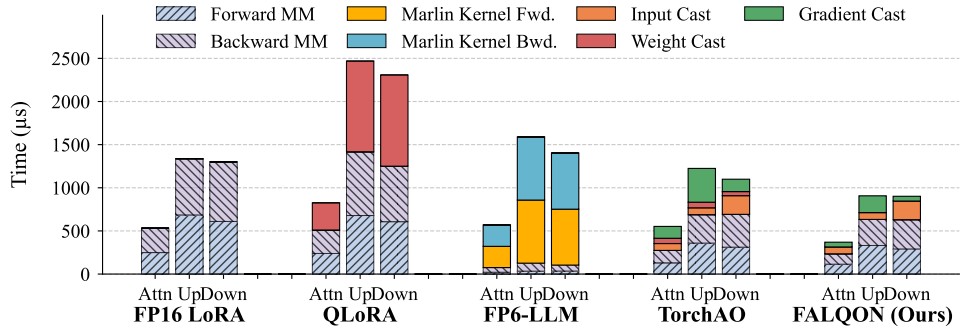

Figure 4: Breakdown analysis of LoRA fine-tuning. Compared to quantized LoRA baselines, which suffer from quantization overheads, FALQON minimizes redundant operations for superior efficiency.

## 6.5 Commonsense reasoning benchmark

Table 3 reports performance on commonsense reasoning tasks. FALQON substantially outperforms the FP6-LLM, reflecting the benefits of its quantization strategy. By contrast, FP6-LLM struggles on tasks such as ARC-C and OBQA, highlighting the difficulties of maintaining accuracy with lower-precision formats. Although some baselines achieve slightly higher averages compared to FALQON, the gap varies by task. For instance, differences on ARC-E and PIQA are within 1–2%, whereas OBQA exhibits a more pronounced gap. Despite not leading on every benchmark, FALQON provides stable results across all tasks and balances accuracy with efficiency.

# 7 Analysis

## 7.1 Breakdown analysis of quantized LoRA fine-tuning

We perform a detailed breakdown analysis to understand computational bottlenecks in quantized LoRA fine-tuning methods, including QLoRA, FP6-LLM, TorchAO, and FALQON. Specifically, we measure the time of forward and backward matrix multiplication, along with the quantization overhead. For FP6-LLM, which uses the Marlin kernel to fuse quantization and computation, we separately measure fused kernel time for forward and backward. The results are shown in Figure 4.

QLoRA introduces significant overhead from weight-only quantization, as frequent dequantization results in nearly double the training time compared to FP16. FP6-LLM partially alleviates quantization overhead through specialized kernels but is constrained by FP16 arithmetic. Similarly, despite TorchAO employing optimized FP8 arithmetic, substantial quantization overhead diminishes overall performance improvements. In contrast, FALQON effectively minimizes quantization overhead by reformulating computational paths and merging adapter computations. This strategy enables FALQON to fully exploit accelerated FP8 matrix multiplication, achieving significantly reduced computation times and superior overall efficiency compared to baseline methods.

## 7.2 Analysis of top-$k$ selection overhead and efficiency

To evaluate the computational cost of the proposed top-$k$ selective update in Algorithm 3, we measure its contribution to the overall training time. Table 4 summarizes the measured latency of each configuration, including the top-$k$ overhead and the resulting reduction in total step time.

The results show that the top-$k$ selection introduces a marginal overhead—less than $0.6\%$ of the total step time—while substantially improving computational efficiency. Since FALQON performs updates only on the most significant components, the arithmetic cost in the update step is considerably reduced. This results in faster iteration times and higher throughput. In practice, the minor additional cost is well compensated by the overall gain in training efficiency, demonstrating that the inclusion of the top-$k$ selection is computationally advantageous. For the effect of top-$k$ on accuracy, please refer to Table 6.

Table 4: Measured overhead and latency from top-$k$ selection.

| Time (ms) | 7B | 13B |
|---|---|---|
| Top-$k$ overhead | +9.97 | +13.85 |
| Step w/o top-$k$ | 1814.98 | 3307.48 |
| Step w/ top-$k$ | 1769.09 | 3210.36 |
| Time reduction | -45.89 | -97.12 |

Table 5: LLaMA-7B Training time, cost, and reduction for LoRA fine-tuning. Financial estimates are based on observed cloud GPU pricing.

| Device | Training Time (days, 8 GPUs) | | | Training Cost ($ USD) | | | Cost Reduction ($ USD) | |
|---|---|---|---|---|---|---|---|---|
| | QLoRA | QA-LoRA | FALQON | QLoRA | QA-LoRA | FALQON | vs QLoRA | vs QA-LoRA |
| RTX 4090 | 89.3 | 153.7 | 35.7 | 6,001 | 10,328 | 1,971 | ⇓4,030 | ⇓8,357 |
| L40S | 98.3 | 164.0 | 37.7 | 35,126 | 58,603 | 10,070 | ⇓25,057 | ⇓48,533 |
| H100 | 31.1 | 25.1 | 13.3 | 41,122 | 33,114 | 13,419 | ⇓27,703 | ⇓19,695 |

## 7.3 Scalability and efficiency of FALQON

We evaluate the scalability of FALQON under multi-adapter workloads representative of large-scale personalization and adaptation tasks [22, 55]. Using the MovieLens-1M dataset [18] with 6,040 users, we estimate training time and monetary cost based on observed rates from cloud GPU platforms: Vast.ai (RTX 4090) [44], AWS G6e (L40S) [37], and AWS P5 (H100) [38]. As shown in Table 5, FALQON achieves substantial cost reductions compared to baselines, owing to up to $3\times$ faster training throughput. These results indicate that FALQON delivers cost-efficient fine-tuning across diverse GPU cards and deployment environments, improving resource efficiency in large-scale adaptation scenarios.

## 7.4 Sensitivity study

We analyze the sensitivity of the LLaMA-7B model to hyperparameter choices, using MMLU accuracy as the evaluation metric. Table 6 illustrates how different learning rates and the number of top-k rows affect performance. In Table 6, we see that reducing the learning rate generally yields slightly improved results. Despite the highest settings (*e.g.*, $2e\text{-}4$ with $k$=10), the overall differences across the tested ranges remain moderate, indicating that our approach is robust to variations in both learning rate and $k$.

Table 6: Performance across different learning rates ($\eta$) and top-$k$ rows

| lr ($\eta$) | Number of Top-$k$ Rows | | | | | |
|---|---|---|---|---|---|---|
| | 1 | 5 | 10 | 20 | 30 | 50 |
| 2e-1 | 0.2465 | 0.2347 | 0.2413 | 0.2295 | 0.2302 | 0.2295 |
| 2e-2 | 0.3209 | 0.3052 | 0.2971 | 0.2933 | 0.2849 | 0.3015 |
| 2e-3 | 0.3410 | 0.3488 | 0.3310 | 0.3460 | 0.3363 | 0.3330 |
| 2e-4 | 0.3460 | 0.3462 | 0.3491 | 0.3470 | 0.3468 | 0.3426 |
| 2e-5 | 0.3454 | 0.3436 | 0.3460 | 0.3465 | 0.3440 | 0.3469 |

In addition, we conduct a sensitivity analysis on batch size and LoRA rank to examine their effects on performance. As shown in the Table 7, performance remains steady across batch sizes (2–16) and ranks (16–128), with metrics ranging narrowly between 0.3418 and 0.3494. Additionally, we analyze the sensitivity of fine-tuning speed to varying batch sizes, with detailed results provided in the Appendix. These results indicate that our method is robust to variations in various hyperparameter settings, demonstrating stable performance without significant tuning requirements. Refer to Section H for further sensitivity studies on the LLaMA-13B model.

Table 7: Sensitivity study on batch size and rank

| Batch | Rank (r) | | | |
|---|---|---|---|---|
| | 16 | 32 | 64 | 128 |
| 2 | 0.3465 | 0.3457 | 0.3473 | 0.3484 |
| 4 | 0.3431 | 0.3494 | 0.3428 | 0.3462 |
| 8 | 0.3418 | 0.3456 | 0.3463 | 0.3482 |
| 16 | 0.3458 | 0.3486 | 0.3491 | 0.3462 |

## 8 Conclusion

We proposed FALQON, a novel FP8 quantization framework for accelerating LoRA fine-tuning. We showed that existing FP8 methods primarily target large-dimensional matrix multiplications and thus experience substantial quantization overhead with small-dimensional LoRA adapters, limiting achievable speedups. To overcome this, FALQON directly merges LoRA adapters into the FP8-quantized backbone and employs a row-wise selective update mechanism, significantly reducing overhead and improving training efficiency. Experimental results demonstrate that FALQON achieves up to a $3\times$ speedup over existing quantized LoRA methods with similar accuracy, providing a practical and efficient solution for fine-tuning large-scale models.

## Acknowledgements

The authors would like to thank Moreh, Inc. for funding this research and inspiring the initial concept.

Jinho Lee is the corresponding author, and he is funded by the following programs: (RS-2024-00395134, RS-2024-00347394, RS-2023-00256081, RS-2021II211343, RS-2025-00564840).

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

## A  Code

We provide the implementation of FALQON in a public GitHub repository: `https://github.com/iamkanghyunchoi/falqon`. Our code is implemented based on the official repository of QLoRA [12]. The repository includes the complete training environment setup, evaluation scripts, and configuration details to ensure reproducibility. The code is under the terms of the MIT license.

## B  Full algorithm of FALQON

This section provides the complete algorithm of the proposed FALQON method. The algorithm integrates LoRA adapters directly into an FP8-quantized backbone from initialization, removing separate adapter computations in forward and backward passes. It also implements a row-wise proxy update mechanism to efficiently propagate significant updates, achieving both computational efficiency and robust training performance. The detailed procedure, including initialization, forward pass computation, gradient calculation, and selective backbone updates, is presented in Algorithm 4.

---

**Algorithm 4** Overall Framework of FALQON

---

1: **Input:** Training data $\{d_i\}_{i=1}^N$, learning rate $\eta$, rank $r$, pretrained backbone $W$, quantization function $Q_{fp8}$, update hyperparameter $k$
2: **Initialization Phase** (Section 5.1)
3: **for** $l = 1, ..., L$ **do**
4:     $\widetilde{W}_l, s_{W_l} \leftarrow Q_{fp8}(W_l)$
5:     $\Delta W_l \leftarrow W_l - \widetilde{W}_l / s_{W_l}$
6:     $\widehat{B}_l, \widehat{A}_l \leftarrow SVD(\Delta W_l, r)$                                   ▷ Melded LoRA (Section 5.1, Equation (10))
7:     $\widetilde{A}_l \leftarrow \lfloor A_l \cdot s_{W_l} \rceil$
8:     $\widetilde{W}' \leftarrow [\widetilde{W}^\top | \widetilde{A}^\top]^\top$                      ▷ $A$ integration (Section 5.2, Equation (12))
9:     $\Delta \text{Buffer}_l \leftarrow \{0\}^{C_{out} \times r}$                                       ▷ $\Delta$Buffer (Section 5.3)
10: **end for**
11: **Training Phase** (Sections 5.2 and 5.3)
12: **for** $i = 1, ..., N$ **do**
13:     $x_0 \leftarrow d_i$
14:     **for** $l = 1, ..., L$ **do**
15:         $\widehat{x}_{l-1}, s_{x_{l-1}} \leftarrow Q_{fp8}(x_{l-1})$
16:         $O_{\text{merged}} \leftarrow \widehat{W}'_l \widehat{x}_{l-1} / s_{W_l} s_{x_{l-1}}$
17:         $O, O_{A_l} \leftarrow O$                                                          ▷ Equation (13)
18:         $x_l \leftarrow O$
19:         $ctx.save\_for\_backward(O_{A_l})$
20:     **end for**
21:     **for** $l = L, ..., 1$ **do**
22:         $O_{A_l} \leftarrow ctx.saved\_tensors$
23:         $\frac{\partial L}{\partial B_l} \leftarrow \frac{\partial L}{\partial x_l} O_{A_l}^T$                                      ▷ Equation (2)
24:         $\Delta \text{Buffer}_l \leftarrow Optimizer(\frac{\partial L}{\partial B_l}, \eta)$
25:         $K \leftarrow topk(\sum_j |\Delta B_{i,j}|; k),$
26:         $\widetilde{W}[K] = \widetilde{W}[K] + \Delta \text{Buffer}[K]A$                                  ▷ Equation (15)
27:     **end for**
28: **end for**

---

Lines 2–10 describe the initialization stage of the FALQON framework. For each linear layer *(line 3)*, weights are first quantized into FP8 format. To get the implicitly embedded LoRA adapters in the FP8-quantized weights, we compute the quantization error *(line 5)*, followed by singular-value decomposition to obtain the low-rank matrices $\widehat{A}$ and $\widehat{B}$ *(line 6)*. Matrix $\widehat{A}$ is quantized *(line 7)* and concatenated directly into the quantized backbone *(line 8)*, and the update buffer $\Delta$Buffer is initialized to zero *(line 9)*.

Table 8: Efficiency analysis across various LoRA ranks. Higher ranks yield improved speed but rapidly increase memory consumption.

| Model | Metric | LoRA Rank | | | | | | | | | | |
|-------|--------|-----|-----|-----|-----|-----|-----|------|------|------|------|-------|
| | | 16 | 32 | 64 | 128 | 256 | 512 | 1024 | 2048 | 4096 | 8192 | 16384 |
| LLaMA-7B | Speed-up ($\times$) | 0.50 | 0.55 | 0.44 | 0.48 | 0.43 | 0.62 | 0.69 | 0.60 | 0.80 | **1.18** | **1.22** |
| | LoRA Mem. (GB) | 0.67 | 0.78 | 0.98 | 1.39 | 2.20 | 3.83 | 7.09 | 13.60 | 26.63 | **52.70** | **104.82** |
| LLaMA-13B | Speed-up ($\times$) | 0.55 | 0.59 | 0.50 | 0.52 | 0.46 | 0.48 | 0.64 | 0.61 | 0.88 | **1.21** | **1.20** |
| | LoRA Mem. (GB) | 1.31 | 1.51 | 1.91 | 2.70 | 4.29 | 7.46 | 13.81 | 26.50 | 51.89 | **102.66** | **204.21** |

Lines 11–28 show the training stage, comprising accelerated forward *(lines 14–20)* and backward computations *(lines 21–27)*. In the forward path, inputs are quantized into FP8 *(line 15)*, then passed through the merged linear layer to obtain both the output $O$ and intermediate result $O_A$ needed for gradient calculations *(line 17)*. The intermediate results are saved for use in backpropagation *(line 19)*. During the backward phase, stored intermediate outputs are loaded *(line 22)* to compute the gradient for matrix $B$ *(line 23)*. An optimizer updates the update buffer $\Delta B$uffer using computed gradients *(line 24)*. Finally, the top-k largest updates in $\Delta B$uffer are selectively applied to the FP8-quantized weights *(lines 25–26)*, ensuring efficient gradient propagation and maintaining training efficiency.

## C  Extended results of speedup comparison across LoRA ranks

Figure 1 in the main text illustrates the efficiency trend of LoRA fine-tuning with varying adapter ranks. While the efficiency appears stagnant in the low-rank region (up to 512), it begins to improve beyond rank 1024, as summarized in Table 8. The results indicate that higher-rank LoRA configurations achieve better computational efficiency but incur substantial memory costs, eventually exceeding those of full fine-tuning.

The observed trend is not counter-intuitive when considering the scaling behavior of quantization and matrix operations. The quantization overhead scales with $O(n^2)$, whereas the matrix multiplication cost scales with $O(n^3)$. Thus, for small-dimensional matrices (common ranks of 16–512), the quantization overhead dominates, masking potential speedups. As the rank increases, the cubic growth of computation amortizes the quadratic quantization cost, leading to improved efficiency beyond rank 1024. However, the accompanying memory footprint increases sharply, making very large ranks (8192–16384) impractical for most fine-tuning setups.

## D  Ablation on matrix $A$ update

In FALQON, updating the matrix $A$ contributes marginally to model quality while incurring additional computational cost. Earlier work, LoRA-FA [54], empirically demonstrated that freezing matrix $A$ has negligible effect on the final fine-tuning performance, while significantly improving memory efficiency and throughput. Building on this observation, we conduct an ablation study to further verify the validity of freezing $A$ within our framework.

Table 9 presents the training time and evaluation accuracy when matrix $A$ is updated or kept frozen. The results confirm that computing the gradient of $A$ not only increases the computational burden—resulting in approximately a 16% slowdown on the 13B model—but also provides minimal gain in model quality. This overhead arises because the gradient of $A$, $\frac{\partial \mathcal{L}}{\partial A} = B^\top \frac{\partial \mathcal{L}}{\partial O} x^\top$, cannot be precomputed. Consequently, freezing $A$ during fine-tuning yields a more efficient optimization process without noticeable degradation in downstream task performance.

## E  Detailed experimental settings

We fine-tune LLaMA-7B and 13B on the Alpaca [42] and OASST1 [23] datasets, then evaluate on the Massively Multitask Language Understanding (MMLU [19]) benchmark, which measures knowledge and reasoning across a diverse set of domains. In addition, we assess our models on HellaSwag [53], PIQA [3], WinoGrande [36], ARC [10], BoolQ [9], and OpenBookQA [30] for commonsense QA, following QA-LoRA's five-shot evaluation protocol via the `lm-eval-harness` framework.

Table 9: Effect of updating matrix *A* on training speed and model quality.

| Model | Method | Time per Step (s) | Alpaca (MMLU) | OASST1 (MMLU) |
|-------|--------|-------------------|---------------|----------------|
| LLaMA-7B | FALQON | 1.80 | 0.3491 | 0.3481 |
| | + Update matrix *A* | 1.94 | 0.3469 | 0.3470 |
| LLaMA-13B | FALQON | 3.26 | 0.4644 | 0.4645 |
| | + Update matrix *A* | 3.89 | 0.4634 | 0.4656 |

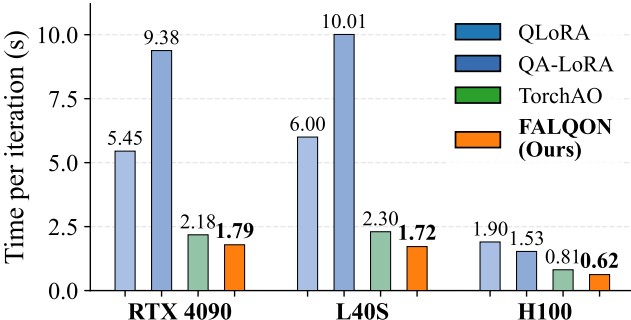

Figure 5: Comparison of training speed on various GPUs.

All computational cost evaluations (Figure 3 and Tables 1a, 1b and 2) and detailed breakdown analyses (Figures 1a, 1b and 4) are conducted on a dedicated node of a local computing cluster. The computing node contains two Intel Xeon Gold 6442Y CPUs (total of 48 physical cores, 96 threads) running at 2.60 GHz, equipped with 1TB DDR5 ECC memory. The node has NVIDIA GeForce RTX 4090 GPUs (24GB VRAM) with NVIDIA driver version 550.54.15 and CUDA 12.4, running on Ubuntu 22.04.4 LTS with Linux kernel 5.15.0-94-generic. Unless otherwise noted, all experiments utilize a single RTX 4090 GPU from this node.

We compare our method (FALQON) against quantized LoRA baselines (QLoRA [12], QA-LoRA [52], IR-QLoRA [35]) and FP quantization methods (FP6-LLM [51], Fishman *et al.* [14] and TorchAO [43]). For each baseline we either use its official implementation or replicate its original experimental setup. In the quantized-LoRA baselines, we adopt weight-only quantization: NF4 for QLoRA and IR-QLoRA, and INT4 for QA-LoRA. For FP6-LLM, we use weight-only FP6-E3M2 and FP6-E2M3 quantization. For FP8-based methods (Fishman *et al.* , TorchAO, and FALQON), we use FP8-E4M3 for weights and activations, and FP8-E5M2 for gradients. We follow the settings of QLoRA: a Paged AdamW optimizer with batch size of 16, max gradient norm of 0.3, learning rate of $2e$-4, $k$=10, and 1,875 training steps. The evaluation is conducted with the trained weight from the end of the training (*i.e.*1,875-th step).

## F  Limitations

In this section, we discuss several limitations of our current approach. While our framework is generally applicable to any neural network layer involving linear operations, its practical effectiveness has thus far only been validated on Decoder-based Transformers such as LLaMA models. Extending our method to additional architectures, including Encoder-based Transformers and Text-to-Image diffusion models, could further demonstrate its broader applicability. However, extending our approach to these additional architectures is beyond the scope of this work, and we leave it as an important direction for future research.

Also, our experimental evaluation primarily considers LLaMA-7B and 13B models, which are sufficiently large to highlight efficiency gains from FP8 quantization, while fitting within typical consumer-level GPUs (VRAM < 24GB). Although evaluating even larger-scale models (e.g., over 70B parameters) would further strengthen our claims regarding scalability and efficiency, this was impractical within our computational constraints. Thus, such assessments remain important directions for future studies when additional computational resources are available.

# G  Speed comparison on various GPUs

We evaluate the fine-tuning speed of quantized LoRA methods (QLoRA, QA-LoRA, TorchAO, and our proposed FALQON) across RTX4090, L40S, and H100 GPUs. The experimental results are shown in Figure 5. The RTX4090 and L40S share NVIDIA's Ada Lovelace architecture, resulting in similar training performance, while the Hopper-based H100 GPU exhibits distinct throughput characteristics. Our proposed method, FALQON, consistently achieves the fastest iteration times across all GPU platforms. Despite the varying relative performances of baseline methods, especially on H100, which is designed for datacenter workload and has a different architecture and high memory bandwidth, FALQON maintains clear superiority, demonstrating its efficiency and adaptability across different hardware environments.

# H  Further experimental results

We present additional MMLU evaluation results in Table 10, extending the analysis in the main body to include both LLaMA-7B and 13B models across Alpaca and OASST1 datasets. For the 13B model, TorchAO and Fishman *et al.* cannot fine-tune due to memory constraints on a single RTX4090 GPU (marked as OOM). Overall, FALQON consistently achieves the fastest training speeds while maintaining competitive or better MMLU scores. These results confirm the effectiveness of FALQON in balancing efficiency, accuracy, and scalability compared to existing FP-quantized methods.

Also, we provide additional experimental results for commonsense reasoning benchmarks in Table 11. While results in the main paper (Table 3) focus on the Alpaca dataset and LLaMA-7B due to space limitations, here we report comprehensive results on both Alpaca and OASST1 datasets across LLaMA-7B and 13B models. Overall, we observe consistent performance trends across datasets and model sizes. Notably, unlike the 7B setting, TorchAO and Fishman *et al.* cannot fine-tune the 13B model on a single RTX4090 GPU due to memory constraints, and thus these cases are noted as out-of-memory (OOM).

In addition, we provide an extended sensitivity study including the LLaMA-13B model (Table 12). The trends observed for the 13B model align closely with those of the 7B counterpart in the main body, with moderate learning rates (around $2 \times 10^{-4}$ to $2 \times 10^{-3}$) generally producing optimal results. However, we notice slightly increased sensitivity at the extreme ends of the tested learning rates, reflecting a mild performance drop at very high ($2e - 1$) or very low ($2e - 5$) settings. Overall, the consistency in results on the moderate learning rate region on 7B and 13B models supports the robustness of our method across varied hyperparameter choices and different model scales.

Lastly, we extend the sensitivity analysis on batch size and LoRA rank, previously discussed for the LLaMA-7B on Alpaca dataset, to additional configurations, datasets, and model scales (Table 13). Results from the OASST1 dataset using the 7B model reveal similar robustness as the Alpaca dataset, with minor fluctuations in performance metrics. For the larger LLaMA-13B models, performance remains consistently stable across varied batch sizes and ranks, suggesting that our method maintains reliability even at larger scales. Collectively, these additional analyses reinforce the general robustness and stability of our method under different hyperparameter choices.

# I  Batch size sensitivity of training speed

We measure training throughput across different batch sizes (Figure 6) and observe that FALQON consistently achieves significantly higher throughput compared to QLoRA and QA-LoRA. Although increasing the batch size naturally reduces throughput due to greater per-step computational load, FALQON demonstrates a substantially smaller decline, showing its superior efficiency and lower computational overhead. This empirically validates that FALQON is especially beneficial for practical scenarios with constrained GPU resources or varying batch size requirements, offering meaningful reductions in overall fine-tuning time.

Table 10: Comparison of low-precision FP quantization methods on Alpaca and OASST1 dataset

| Model | Method | Type | Time / Step (s) | # Trainable Params | Alpaca (MMLU) | | | | | OASST1 (MMLU) | | | | |
|---|---|---|---|---|---|---|---|---|---|---|---|---|---|---|
| | | | | | Hum. | STEM | Social | Other | Avg. | Hum. | STEM | Social | Other | Avg. |
| 7B | TorchAO | FP8 | 2.18 | 160M | 0.3231 | 0.2969 | 0.3679 | 0.3785 | 0.3393 | 0.3273 | 0.3092 | 0.3672 | 0.3869 | 0.3452 |
| | FP6-LLM | E2M3 | 2.72 | 160M | 0.2421 | 0.2125 | 0.2171 | 0.2398 | 0.2295 | 0.2448 | 0.2125 | 0.2177 | 0.2411 | 0.2308 |
| | FP6-LLM | E3M2 | 2.72 | 160M | 0.2487 | 0.2693 | 0.2532 | 0.2333 | 0.2509 | 0.2423 | 0.2249 | 0.2190 | 0.2411 | 0.2330 |
| | Fishman et al. [14] | FP8 | 2.29 | 160M | 0.3337 | 0.3108 | 0.3893 | 0.3923 | 0.3537 | 0.3241 | 0.2969 | 0.3773 | 0.3714 | 0.3401 |
| | FALQON (Ours) | FP8 | **1.79** | 80M | 0.3322 | 0.3086 | 0.3858 | 0.3795 | 0.3491 | 0.3373 | 0.3130 | 0.3776 | 0.3708 | 0.3481 |
| 13B | TorchAO | FP8 | - | 160M | OOM | OOM | OOM | OOM | OOM | OOM | OOM | OOM | OOM | OOM |
| | FP6-LLM | E2M3 | 10.19 | 160M | 0.2421 | 0.2141 | 0.2171 | 0.2398 | 0.2298 | 0.2640 | 0.2518 | 0.2424 | 0.2626 | 0.2562 |
| | FP6-LLM | E3M2 | 10.19 | 160M | 0.2425 | 0.2211 | 0.2236 | 0.2417 | 0.2334 | 0.2425 | 0.2119 | 0.2190 | 0.2401 | 0.2300 |
| | Fishman et al. [14] | FP8 | - | 160M | OOM | OOM | OOM | OOM | OOM | OOM | OOM | OOM | OOM | OOM |
| | FALQON (Ours) | FP8 | 3.24 | 80M | 0.4408 | 0.3578 | 0.4774 | 0.4799 | 0.4644 | 0.4436 | 0.3571 | 0.4771 | 0.4754 | 0.4645 |

Table 11: Comparison on common sense reasoning benchmarks

| Dataset | Model | Method | Benchmark Results | | | | | | | |
|---|---|---|---|---|---|---|---|---|---|---|
| | | | ARC-C | ARC-E | BoolQ | HellaSwag | OBQA | PIQA | Winogrande | Average |
| Alpaca | 7B | QLoRA | 0.4735 | 0.7891 | 0.7856 | 0.5787 | 0.3660 | 0.7976 | 0.7159 | 0.6438 |
| | | QA-LoRA | 0.4735 | 0.7723 | 0.7511 | 0.5618 | 0.3620 | 0.7867 | 0.7230 | 0.6329 |
| | | IR-QLoRA | 0.4812 | 0.7786 | 0.7902 | 0.5819 | 0.3680 | 0.8009 | 0.7230 | 0.6463 |
| | | FP6-LLM ($E2M3$) | 0.2125 | 0.2681 | 0.3783 | 0.2616 | 0.1400 | 0.5229 | 0.5075 | 0.3273 |
| | | FP6-LLM ($E3M2$) | 0.2073 | 0.2647 | 0.3783 | 0.2600 | 0.1020 | 0.5239 | 0.4957 | 0.3188 |
| | | TorchAO (FP8) | 0.4753 | 0.7870 | 0.7933 | 0.5824 | 0.3640 | 0.7960 | 0.7206 | 0.6455 |
| | | Fishman et al. (FP8) | 0.4804 | 0.7896 | 0.7792 | 0.5826 | 0.3640 | 0.7949 | 0.7222 | 0.6447 |
| | | FALQON (Ours) | 0.4608 | 0.7786 | 0.7676 | 0.5711 | 0.3420 | 0.7905 | 0.7135 | 0.6320 |
| Alpaca | 13B | QLoRA | 0.5299 | 0.8161 | 0.8269 | 0.6087 | 0.3700 | 0.8085 | 0.7640 | 0.6749 |
| | | QA-LoRA | 0.5179 | 0.8051 | 0.8064 | 0.5930 | 0.3580 | 0.8074 | 0.7719 | 0.6657 |
| | | IR-QLoRA | 0.5247 | 0.8110 | 0.8407 | 0.6064 | 0.3640 | 0.8118 | 0.7537 | 0.6732 |
| | | FP6-LLM ($E2M3$) | 0.2048 | 0.2731 | 0.3783 | 0.2609 | 0.1340 | 0.5283 | 0.4949 | 0.3249 |
| | | FP6-LLM ($E3M2$) | 0.2133 | 0.2685 | 0.3783 | 0.2605 | 0.1440 | 0.5310 | 0.5075 | 0.3290 |
| | | TorchAO (FP8) | OOM | OOM | OOM | OOM | OOM | OOM | OOM | OOM |
| | | Fishman et al. (FP8) | OOM | OOM | OOM | OOM | OOM | OOM | OOM | OOM |
| | | FALQON (Ours) | 0.5179 | 0.8064 | 0.7887 | 0.6010 | 0.3560 | 0.8030 | 0.7648 | 0.6625 |
| OASST1 | 7B | QLoRA | 0.4650 | 0.7849 | 0.7813 | 0.5921 | 0.3640 | 0.7900 | 0.7222 | 0.6428 |
| | | QA-LoRA | 0.4727 | 0.7765 | 0.7731 | 0.5816 | 0.3700 | 0.7954 | 0.7167 | 0.6409 |
| | | IR-QLoRA | 0.4761 | 0.7715 | 0.7991 | 0.5956 | 0.3780 | 0.7889 | 0.7072 | 0.6452 |
| | | FP6-LLM ($E2M3$) | 0.2031 | 0.2614 | 0.6162 | 0.2581 | 0.1280 | 0.5234 | 0.4862 | 0.3538 |
| | | FP6-LLM ($E3M2$) | 0.1911 | 0.2736 | 0.3783 | 0.2599 | 0.1260 | 0.5408 | 0.4949 | 0.3235 |
| | | TorchAO (FP8) | 0.4727 | 0.7854 | 0.7991 | 0.5932 | 0.3640 | 0.7911 | 0.7206 | 0.6466 |
| | | Fishman et al. (FP8) | 0.4804 | 0.7875 | 0.7911 | 0.5966 | 0.3620 | 0.7922 | 0.7222 | 0.6474 |
| | | FALQON (Ours) | 0.4514 | 0.7769 | 0.7703 | 0.5716 | 0.3400 | 0.7889 | 0.7174 | 0.6309 |
| OASST1 | 13B | QLoRA | 0.5213 | 0.8110 | 0.8110 | 0.6170 | 0.3520 | 0.8074 | 0.7672 | 0.6696 |
| | | QA-LoRA | 0.5341 | 0.8114 | 0.8193 | 0.6178 | 0.3560 | 0.8107 | 0.7743 | 0.6748 |
| | | IR-QLoRA | 0.5307 | 0.8026 | 0.7963 | 0.6227 | 0.3660 | 0.7965 | 0.7616 | 0.6681 |
| | | FP6-LLM ($E2M3$) | 0.2585 | 0.3413 | 0.5024 | 0.3266 | 0.1580 | 0.5778 | 0.6409 | 0.4008 |
| | | FP6-LLM ($E3M2$) | 0.2065 | 0.2702 | 0.4550 | 0.2589 | 0.1180 | 0.5261 | 0.4791 | 0.3305 |
| | | TorchAO (FP8) | OOM | OOM | OOM | OOM | OOM | OOM | OOM | OOM |
| | | Fishman et al. (FP8) | OOM | OOM | OOM | OOM | OOM | OOM | OOM | OOM |
| | | FALQON (Ours) | 0.5154 | 0.8030 | 0.7902 | 0.6012 | 0.3460 | 0.8079 | 0.7743 | 0.6626 |

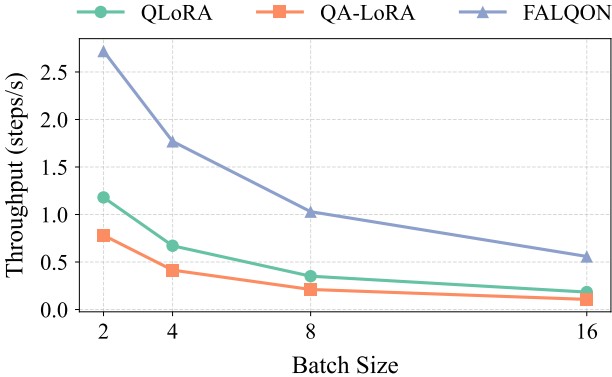

Figure 6: Throughput at varying batch sizes

Table 12: Performance across different learning rates ($\eta$) and top-$k$ rows for Alpaca and OASST1 datasets

| Dataset | LLaMA | lr ($\eta$) | Number of Top-$k$ Rows | | | | | |
|---|---|---|---|---|---|---|---|---|
| | | | 1 | 5 | 10 | 20 | 30 | 50 |
| Alpaca | 7B | 2e-1 | 0.2465 | 0.2347 | 0.2413 | 0.2295 | 0.2302 | 0.2295 |
| | | 2e-2 | 0.3209 | 0.3052 | 0.2971 | 0.2933 | 0.2849 | 0.3015 |
| | | 2e-3 | 0.3410 | 0.3488 | 0.3310 | 0.3460 | 0.3363 | 0.3330 |
| | | 2e-4 | 0.3460 | 0.3462 | 0.3491 | 0.3470 | 0.3468 | 0.3426 |
| | | 2e-5 | 0.3454 | 0.3436 | 0.3460 | 0.3465 | 0.3440 | 0.3469 |
| | 13B | 2e-1 | 0.3047 | 0.2300 | 0.2322 | 0.2295 | 0.2295 | 0.2294 |
| | | 2e-2 | 0.4507 | 0.4591 | 0.4472 | 0.4394 | 0.3820 | 0.3774 |
| | | 2e-3 | 0.4648 | 0.4694 | 0.4634 | 0.4629 | 0.3924 | 0.4160 |
| | | 2e-4 | 0.4638 | 0.4657 | 0.4644 | 0.4650 | 0.4301 | 0.4294 |
| | | 2e-5 | 0.4273 | 0.4270 | 0.4284 | 0.4298 | 0.4269 | 0.4290 |
| OASST1 | 7B | 2e-1 | 0.5509 | 0.3931 | 0.3570 | 0.2697 | 0.2571 | 0.2605 |
| | | 2e-2 | 0.3544 | 0.3627 | 0.3494 | 0.3475 | 0.5799 | 0.3400 |
| | | 2e-3 | 0.3484 | 0.3483 | 0.3495 | 0.3434 | 0.5855 | 0.3504 |
| | | 2e-4 | 0.3452 | 0.3484 | 0.3481 | 0.3462 | 0.5732 | 0.3472 |
| | | 2e-5 | 0.3468 | 0.3493 | 0.3462 | 0.3458 | 0.5721 | 0.3454 |
| | 13B | 2e-1 | 0.3446 | 0.2632 | 0.2404 | 0.2352 | 0.2295 | 0.2423 |
| | | 2e-2 | 0.4580 | 0.4611 | 0.4611 | 0.4586 | 0.4519 | 0.4358 |
| | | 2e-3 | 0.4637 | 0.4616 | 0.4578 | 0.4574 | 0.4566 | 0.4514 |
| | | 2e-4 | 0.4645 | 0.4651 | 0.4645 | 0.4662 | 0.4624 | 0.4645 |
| | | 2e-5 | 0.4671 | 0.4654 | 0.4655 | 0.4656 | 0.4647 | 0.4659 |

Table 13: Sensitivity study on batch size and rank

| Dataset | Model | Batch | Rank (r) | | | |
|---|---|---|---|---|---|---|
| | | | 16 | 32 | 64 | 128 |
| Alpaca | 7B | 2 | 0.3465 | 0.3457 | 0.3473 | 0.3484 |
| | | 4 | 0.3431 | 0.3494 | 0.3428 | 0.3462 |
| | | 8 | 0.3418 | 0.3456 | 0.3463 | 0.3482 |
| | | 16 | 0.3458 | 0.3486 | 0.3491 | 0.3462 |
| | 13B | 2 | 0.4657 | 0.4662 | 0.4641 | 0.4668 |
| | | 4 | 0.4650 | 0.4626 | 0.4640 | 0.4654 |
| | | 8 | 0.4649 | 0.4634 | 0.4656 | 0.4643 |
| | | 16 | 0.4647 | 0.4647 | 0.4644 | 0.4648 |
| OASST1 | 7B | 2 | 0.3477 | 0.3467 | 0.3490 | 0.3502 |
| | | 4 | 0.3439 | 0.3442 | 0.3461 | 0.3463 |
| | | 8 | 0.3469 | 0.3476 | 0.3464 | 0.3448 |
| | | 16 | 0.3460 | 0.3482 | 0.3481 | 0.3467 |
| | 13B | 2 | 0.4658 | 0.4647 | 0.4635 | 0.4640 |
| | | 4 | 0.4662 | 0.4643 | 0.4646 | 0.4639 |
| | | 8 | 0.4624 | 0.4646 | 0.4644 | 0.4652 |
| | | 16 | 0.4637 | 0.4673 | 0.4645 | 0.4638 |

