# OpenReview forum: "FALQON: Accelerating LoRA Fine-tuning with Low-Bit Floating-Point Arithmetic"
_NeurIPS.cc/2025/Conference — NeurIPS 2025 poster_

### Official Review · Reviewer_UHn5 · 2025-06-30

**Clarity:** 1
**Significance:** 2
**Originality:** 3
**Rating:** 4
**Confidence:** 5

**Summary:**

This paper presents FALQON, a novel framework designed to accelerate LoRA fine-tuning for large language models by leveraging FP8 arithmetic. The key insight is that while FP8 quantization accelerates large matrix multiplications, it incurs significant overhead for small matrices like those used in LoRA, often negating performance gains. To address this, the authors propose a method to merge LoRA adapters directly into the FP8-quantized backbone weights, thus avoiding the computational cost of separate adapter paths. They introduce a melded LoRA formulation, which repurposes quantization error as a low-rank approximation, and develop an efficient gradient computation strategy that does not require explicitly storing adapter parameters. Moreover, a row-wise proxy update mechanism is introduced to selectively update meaningful components of the quantized backbone, further enhancing efficiency. Experimental results show that FALQON achieves up to 3× speedup compared to existing quantized LoRA approaches while maintaining comparable accuracy across multiple benchmarks.

**Questions:**

Motivation for FP8:
- The paper argues that quantized LoRA methods have substantial overhead but does not fully justify the use of FP8 specifically for LoRA. Can the authors elaborate on the real-world scenarios or hardware settings where such speed gains justify potential accuracy losses or increased complexity?

Preliminary Analysis Scope:
- The discussion on when FP8 becomes beneficial (e.g., matrix size ≥ 4K) is theoretical. Could the authors include experimental data showing the turning point where FP8 outperforms FP16 in practice for LoRA modules? This would better inform practitioners when FP8 should be preferred.

Clarification on Table 1 Results:

 - Is QLoRA implemented as in the original paper with 4-bit NF4 quantization?

 - Why are QA-LoRA and IR-QLoRA slower than QLoRA despite having fewer trainable parameters, especially when QA-LoRA claims to outperform QLoRA in speed in its original paper?

Loss Curve Comparisons:
- Could the authors provide training loss curves for all quantized LoRA baselines compared to FALQON? This would help verify convergence behaviors, training stability, and potential trade-offs between speed and optimization quality.

**Ethical Concerns:**

["NO or VERY MINOR ethics concerns only"]

**Final Justification:**

Addressed my concerns

**Limitations:**

yes

**Quality:**

2

**Strengths And Weaknesses:**

This paper introduces a technically solid framework, FALQON, to accelerate LoRA fine-tuning using FP8 arithmetic. The design is clear and thoughtful, particularly the use of quantization error as a low-rank approximation and the integration of LoRA adapters into the quantized backbone. The melded LoRA formulation and row-wise proxy update mechanism are original contributions that show strong potential to reduce overhead while preserving accuracy. The paper is clearly written, with well-structured explanations and informative figures that effectively convey the technical design and empirical results. Its contributions are timely and relevant, addressing the under-explored challenge of improving computational efficiency in parameter-efficient fine-tuning, which enhances its significance.

However, the motivation to adopt FP8 training specifically for LoRA is not fully convincing — it remains unclear whether the potential accuracy trade-offs are justified across practical use cases. Furthermore, some inconsistencies in the experimental results raise questions. For instance, QA-LoRA and IR-QLoRA, which have fewer trainable parameters than QLoRA, appear slower in Table 1, contradicting their original papers. Clarification on these implementation details is necessary to assess the fairness and validity of the comparisons.

---

> ### Author Rebuttal · Authors · 2025-07-31
>
> We appreciate the valuable comments from the reviewer. We also thank the reviewer for acknowledging our paper to be technically solid, the design is clear and thoughtful, and the paper is clearly written and the contribution is timely.
>
> This rebuttal addresses the reviewer’s concerns and suggestions. Below is a brief summary of this rebuttal.
>
> * **W1, Q1-Motivation for FP8 is not fully convincing**: We elaborate on the real-world scenarios (4 tasks from 6 papers) in the paper (line 111-118).
> * **W2, Q3.2-Why are QA-LoRA and IR-QLoRA slower than QLoRA despite having fewer trainable parameters, especially when QA-LoRA claims to outperform QLoRA in speed in its original paper?**: The original claim is about inference speed. Our paper focuses on fine-tuning speed, which is determined by the computational overhead rather than the number of trainable parameters.
> * **Q2-Could the authors include experimental data showing the turning point where FP8 outperforms FP16 in practice for LoRA modules?**: We present the experimental results and the turning point is around rank 8192.
> * **Q3.1-Is QLoRA implemented as in the original paper with 4-bit NF4 quantization?**: Yes.
> * **Q4-Loss curve comparisons**: We provide a training loss table of every hundred steps instead of a figure, due to the limitation of the conference policy.
>
> We will improve our manuscript to address the reviewer. Please refer to the following for the detailed rebuttal.
>
> ***
> ### **W1, Q1: Can the authors elaborate on the real-world scenarios or hardware settings where such speed gains justify potential accuracy losses or increased complexity?**
>
> $\to$ For this, we can refer to the real-world scenarios provided in the paper (line 111-118), which trains up to millions of LoRA adapters, and therefore the speed up is especially beneficial. In lines 111-118, we present four scenarios that train a large amount of LoRA adapters for *personalization [54, 50, 26], multitask learning [44], domain adaptation [11], and multilingual summarization*. For example, PLoRA [54] assumes millions of users, and iLoRA [C1] conducts evaluation on a few hundred to 6 million users. Therefore,  we believe our framework significantly provides benefits to these scenarios.
>
> [C1] Kong, Xiaoyu, et al. "Customizing language models with instance-wise lora for sequential recommendation." NeurIPS (2024).
>
> ### **W2, Q3.2: QA-LoRA and IR-QLoRA, which have fewer trainable parameters than QLoRA, appear slower in Table 1, contradicting their original papers. Why are QA-LoRA and IR-QLoRA slower than QLoRA despite having fewer trainable parameters, especially when QA-LoRA claims to outperform QLoRA in speed in its original paper?**
>
> $\to$ We respectfully disagree that our results contradict the baselines’ original papers. The QA-LoRA claims that they are faster in the **inference** stage, while we report the **fine-tuning** speed.
>
> In the original QA-LoRA paper, they claimed as *“The major advantage of QA-LoRA, compared to QLoRA, lies in the **inference** stage where it is faster.”* on page 5. This clearly shows that they mentioned the inference stage. However, in our target scenario, the fine-tuning speed is primarily determined by computational overhead, rather than the number of trainable parameters. As QA-LoRA and IR-QLoRA utilize auxiliary computations in addition to QLoRA, they are slower than QLoRA in **fine-tuning**, but they are faster in the **inference stage** due to optimization focused on inference efficiency. We believe this does not contradict the original paper, but aligns with their proposed algorithms.
>
> ### **Q2: Could the authors include experimental data showing the turning point where FP8 outperforms FP16 in practice for LoRA modules?**
>
> $\to$ The turning point of the LoRA adapter is rank 8192, which is an exceptionally large rank compared to common settings. We provide a detailed analysis across LoRA rank in Table R1.
>
> **Table R1:** Efficiency analysis across various LoRA ranks
> | |Rank:|16|32|64|128|256|512|1024|2048|4096|8192|16384|
> |---|---|---|---|---|---|---|---|---|---|---|---|---|
> |LLaMA-7B|Speed Up|0.50$\times$|0.55$\times$|0.44$\times$|0.48$\times$|0.43$\times$|0.62$\times$|0.69$\times$|0.60$\times$|0.80$\times$|**1.18$\times$**|**1.22$\times$**|
> |FT: 49.01GB|LoRA Mem.|0.67*GB*|0.78*GB*|0.98*GB*|1.39*GB*|2.20*GB*|3.83*GB*|7.09*GB*|13.60*GB*|26.63*GB*|**52.70*GB***|**104.82*GB***|
> |LLaMA-13B|Speed Up|0.55$\times$|0.59$\times$|0.50$\times$|0.52$\times$|0.46$\times$|0.48$\times$|0.64$\times$|0.61$\times$|0.88$\times$|**1.21$\times$**|**1.20$\times$**|
> |FT: 95.48*GB*|LoRA Mem.|1.31*GB*|1.51*GB*|1.91*GB*|2.70*GB*|4.29*GB*|7.46*GB*|13.81*GB*|26.50*GB*|51.89*GB*|**102.66*GB***|**204.21*GB***|
> ||
>
> The experiments compare the computation time of LoRA adapters on FP16 and FP8 arithmetic. In addition, we provide the memory consumption of full fine-tuning (FT) and LoRA adapter to show the efficiency in relation to LoRA rank. Even if FP8 computation is beneficial in ranks 8192 and 16384, the memory consumption on those ranks exceeds that of full fine-tuning, making LoRA inefficient.
>
> ### **Q3.1: Is QLoRA implemented as in the original paper with 4-bit NF4 quantization?**
>
> $\to$ Yes, we used the NF4 quantization and also used the original code provided by the QLoRA’s authors.
>
> ### **Q4: Could the authors provide training loss curves for all quantized LoRA baselines compared to FALQON?**
>
> $\to$ Unfortunately, we could not present the figures due to the NeurIPS conference policy this year. Therefore, we provide a table showing the average training loss of every hundred steps (Table R2).
>
> **Table R2:** Loss comparison across training steps
> |Steps|100|200|300|400|500|600|700|800|900|1000|1100|1200|1300|1400|1500|1600|1700|1800|
> |---|---|---|---|---|---|---|---|---|---|---|---|---|---|---|---|---|---|---|
> |FP16|4.379|4.290|4.337|4.361|4.235|4.326|4.244|4.160|4.194|4.341|4.227|3.971|4.156|4.194|4.203|4.156|4.047|4.003|
> |QLoRA|4.397|4.302|4.347|4.371|4.244|4.335|4.252|4.169|4.202|4.349|4.235|3.979|4.164|4.202|4.211|4.164|4.055|4.009|
> |TorchAO|4.382|4.293|4.339|4.364|4.237|4.328|4.246|4.162|4.196|4.343|4.230|3.975|4.161|4.198|4.207|4.160|4.054|4.012|
> |Fishmanetal.|4.381|4.291|4.338|4.363|4.236|4.327|4.245|4.161|4.195|4.342|4.229|3.974|4.159|4.197|4.205|4.158|4.051|4.008|
> |FALQON|5.009|4.653|4.544|4.393|4.361|4.368|4.405|4.378|4.406|4.476|4.387|4.346|4.367|4.418|4.384|4.467|4.329|4.378|
> ||
>
> Overall, the results show the training curve of each method. The magnitude of the loss is similar to non-quantized FP16 LoRA, and the loss is reduced as the training continues. Also, please note that the training loss is not directly related to the performance of the downstream task (e.g., MMLU accuracy), due to the nature of the language models. Therefore, a larger training loss does not necessarily mean lower quality of the model.
>
> ***
> We thank the reviewer for the helpful comments. We believe that our rebuttal well addresses the reviewer’s concern and potentially clarifies our contribution. We welcome further comments and would be happy to discuss them.

---

> ### Comment · Reviewer_UHn5 · 2025-08-05
> **Response**
>
> I really appreciate your thoughtful and thorough comments, which have resolved many of my concerns.
> However, one aspect remains unclear to me. Although QA-LoRA primarily focuses on inference speed, Table 2 in the QA-LoRA paper also reports that its fine-tuning time is considerably shorter than QLoRA. Could you clarify this point?

---

> > ### Author Response · Authors · 2025-08-05
> >
> > We appreciate the reviewer's valuable comment. We checked Table 2 from the original QA-LoRA paper, confirming it indeed compares fine-tuning speed.
> >
> >
> > To verify these results, we freshly cloned the official GitHub repositories for both QLoRA and QA-LoRA and ran the experiments with the original configurations. Our results consistently showed QLoRA is faster than QA-LoRA, in contrast to what was originally reported in QA-LoRA.
> >
> >
> > In addition to our results, recent papers [C2, C3] also reported that QLoRA is faster than QA-LoRA. The fine-tuning times from these papers are summarized in Table R3 below:
> >
> >
> > | Model      | Method    | QEFT [C2] | L4Q [C3] | FALQON (Ours) |
> > |------------|-----------|:-----------:|:----------:|:---------------:|
> > | **7 B**    | QLoRA     | 3.5 h     | 9.9 h    | 2.84 h        |
> > |            | QA-LoRA   | 5.2 h     | 11.2 h   | 4.92 h        |
> > ||
> > | **13 B**   | QLoRA     | 6.1 h     | 18.0 h   | 4.88 h        |
> > |            | QA-LoRA   | 8.9 h     | 19.8 h   | 9.39 h        |
> > ||
> >
> > These results align with our observations that QA-LoRA is consistently slower than QLoRA. Please note that the direct comparison of fine-tuning time across papers is not possible, due to different experimental settings.
> >
> > We suspect the different trend is related to the quantization libraries used. QLoRA utilizes NF4 quantization from the bitsandbytes library, while QA-LoRA employs INT4 quantization from the GPTQ library. According to user reports (e.g., GitHub issues) around early 2023—when QA-LoRA was published—the bitsandbytes library had performance issues. This may explain why QA-LoRA originally appeared faster, as the original paper attributes its speed to CUDA optimizations. After two years of library updates, CUDA optimizations, and new GPU architectures, we believe the relative performance between these methods has changed.
> >
> > We hope this addresses the reviewer’s concern. We again thank the reviewer for the discussion.
> >
> > [C2] Lee, Changhun, et al. "QEFT: Quantization for Efficient Fine-Tuning of LLMs." Findings of the Association for Computational Linguistics: EMNLP 2024. 2024.
> >
> > [C3] Jeon, Hyesung, et al. "L4Q: Parameter Efficient Quantization-Aware Fine-Tuning on Large Language Models." arXiv preprint arXiv:2402.04902 (2024).

---

> > > ### Comment · Reviewer_UHn5 · 2025-08-05
> > >
> > > Thank you for the detailed response and for referencing other works that show similar trends. Your explanation seems reasonable, and I appreciate the clarification. I will adjust my score accordingly.

---

> > > > ### Author Response · Authors · 2025-08-06
> > > >
> > > > We greatly appreciate the reviewer’s comments and engagement throughout the discussion. We are delighted that all concerns have been addressed and welcome the positive feedback. We will revise our paper accordingly.

---

### Official Review · Reviewer_EcqX · 2025-06-30

**Clarity:** 2
**Significance:** 3
**Originality:** 2
**Rating:** 4
**Confidence:** 3

**Summary:**

The work suggests merging LoRA adapters into an FP8 quantized backbone to speed up quantized fine tuning with FP8. Instead of running a separate FP16 path for the two LoRA multiplications in $BAx$ the factor $\hat{A}$ is folded into the FP8 matrix so a single operations produces both of the multiplications with $x$, $\widetilde{W}x$ and the LoRA factor $\hat{A}x$.
This means that gradients are only needed for $\hat{B}$ and conversions between FP16 and FP8 are limited, providing a computational speedup. Experiments confirm wall time speedup and not large deviation from task evaluations.

**Questions:**

* What is special about FP8? Compared to say, INT4/NF4?
* Section 4 / Figure 1: What is behind the dashed lines, theoretical maximum, and pre-training speed up? What is the empirical setup? Do you take random matrices of random dimensions? This is not clear.
* The sign of the error term flips from equation 8 to 9, keep it as negation?
* Line 184-185: I don’t see another interpretation as possible, what do you mean?

**Ethical Concerns:**

["NO or VERY MINOR ethics concerns only"]

**Final Justification:**

I leave my score at 4 since I am not sure about the broader impact of the work, the "trick" developed is nice though.

**Limitations:**

Yes

**Quality:**

3

**Strengths And Weaknesses:**

Strengths:
* The approach makes sense, removing these redundant operations is valuable (and could be implemented with a fused kernel for larger improvements?)
* The low rank treatment has been used before LoftQ, LQLoRA, LoQT, initialising with SVD is empirically sound
* Cutting memory and compute is important
* Experimental comparisons and ablations are well rounded, good to see both performance and speed checked

Weaknesses
* Would have been good to have some sort of full picture overview or the key derivations together to get a concise overview, either with equations, a diagram or a combination of the two. I found it quite hard to follow. I think a more pedagogical approach could have been taken. The contribution is hidden away until page 5!
* Writing and grammar can be improved. Some sentences are very long and logical flow is not always evident, e.g. line 92. Other sentences are vague, e.g. line 149.
* You say in the introduction that FP8 quantization overhead outweighs the benefits from FP8 multiplication. I think you can make it clearer it is the speed and not the memory you refer to
* Same in claims about cost of FP8<->FP16 in line 104-105. You say there will be a detailed analysis in section 4, but it is somewhat preliminary (as the section title says).
* In the background on LoRA: you may want to mention that the adapters can be downcast.
* In some places it would be good to make it clear you are talking about computational speed and not memory as is often the focus of the low rank papers.
* I would say what your method does earlier, at least at a high level. You keep mentioning it is better (which gets repetitive), but only at page 5 do you say how it works!
* Typo: Line 84, you say $\widetilde{W}$ is high precision
* Figure 2: For this to be easier to read it might be good to include the operations (addition and multiplication)
* Equation 9 uses the same approach as LoftQ and LQLoRA to initialize the quantization error, if I'm not mistaken, maybe worth mentioning both
* No error bars, you could have at least bootstrapped intervals for the evaluations
* Note on code: it’s 244MB uncompressed! That seems very excessive. Also, not all .git folders were deleted.

---

> ### Author Rebuttal · Authors · 2025-07-31
>
> We genuinely appreciate the reviewer for the helpful comments and feedback. We are happy that the reviewer finds the proposed methods are valuable and sound.
>
> We present this rebuttal to address the reviewer’s concerns and questions. Here is a brief summary of the contents.
>
> * **W1, W9-Would have been good to have a concise overview**: We clarified the algorithmic overview and improved the explanation of key derivations, including diagram updates and explicit operation descriptions.
> * **W4-Request for detailed analysis for matrix dimension**: We provide a detailed speed-up analysis across matrix sizes, showing that FP8 achieves empirical acceleration for dimensions above 4K.
> * **W11-Error bar for evaluations**: We provide two-sigma error bars for the experiments, confirming consistency.
> * **Q1-What is special about FP8 compared to INT4/NF4?**: We clarify that FP8 uniquely accelerates fine-tuning via fast GPU arithmetic, while INT4/NF4 primarily reduce inference memory.
> * **Typos and Writing suggestions**: We have revised and clarified key sentences as suggested.
>
> Here, we provide a detailed rebuttal.
>
> ***
> ### **W1, W9: Would have been good to have some sort of full picture overview or the key derivations together to get a concise overview. I think a more pedagogical approach could have been taken. Figure 2: For this to be easier to read it might be good to include the operations (addition and multiplication)**
>
> $\to$ Thank you for the suggestion! Please note that we have included the complete algorithm overview in Appendix Section B. Due to the page limits, we provided partial algorithms for each of the proposed methods in the main body of the paper.
>
> Regarding Figure 2, we will add a modified version with the operations in the manuscript. Unfortunately, we could not post the revised figure in rebuttal, due to the system’s limitations. Therefore, we briefly mention how the change will happen. In Figure 2, activations ($\tilde X$ and $O$) connected to the weights ($\tilde W$, $\tilde A$, and $\tilde B$) with arrows are matrix multiplication. Weights and activations connected to (Quant) are quantization operations. Bold arrow in 3. Update figure is an addition for the weight update.
>
>
> ### **W2: Some sentences are very long or vague.**
>
> $\to$ Thank you for the constructive feedback. We will improve the sentences as follows:
>
> * Line 92 $\Rightarrow$ Low-precision FP8 formats (e.g., E4M3 or E5M2) have a narrow representable range. Therefore, a higher-precision tensor $X$ (e.g., FP32 or FP16) must be scaled by a per-tensor scaling factor $s_X$. The scaled tensor $s_X \cdot X$ is then rounded to FP8, as follows:
>
> * Line 149 $\Rightarrow$ LoRA fine-tuning commonly involves small-dimensional matrices. In such cases, the computational overhead of quantization outweighs the speedup.
>
> ### **W3 and W6: It would be good to make it clear you are talking about computational speed and not memory, as is often the focus of the low rank papers.**
>
> $\to$ We thank the reviewer for the great suggestion! We will further clarify this throughout the manuscript. For example, we will add “Unlike most of the prior LoRA-based methods that aim to reduce the memory cost, our method aims to speedup the fine-tuning of LoRA by leveraging the FP8 arithmetics, suggesting an algorithmic approach to maintain the accuracy while maximizing the speed.”
>
> ### **W4: Claims about cost of FP8<->FP16 in line 104-105. You say there will be a detailed analysis in section 4, but it is somewhat preliminary.**
>
> $\to$ We further provide a detailed analysis regarding the FP8 efficiency across LoRA rank in Table R1, comparing computation time and memory usage with FP16.
>
> **Table R1:** Efficiency analysis across various LoRA ranks
> | |Rank:|16|32|64|128|256|512|1024|2048|4096|8192|16384|
> |---|---|---|---|---|---|---|---|---|---|---|---|---|
> |LLaMA-7B|Speed Up|0.50x|0.55x|0.44x|0.48x|0.43x|0.62x|0.69x|0.60x|0.80x|**1.18x**|**1.22x**|
> |FT: 49.01GB|LoRA Mem.|0.67*GB*|0.78*GB*|0.98*GB*|1.39*GB*|2.20*GB*|3.83*GB*|7.09*GB*|13.60*GB*|26.63*GB*|**52.70*GB***|**104.82*GB***|
> |LLaMA-13B|Speed Up|0.55x|0.59x|0.50x|0.52x|0.46x|0.48x|0.64x|0.61x|0.88x|**1.21x**|**1.20x**|
> |FT: 95.48*GB*|LoRA Mem.|1.31*GB*|1.51*GB*|1.91*GB*|2.70*GB*|4.29*GB*|7.46*GB*|13.81*GB*|26.50*GB*|51.89*GB*|**102.66*GB***|**204.21*GB***|
> ||
>
> The results indicate that FP8 incurs a slowdown for typical LoRA ranks under 4K, and only becomes faster than FP16 at extremely high ranks (over 8192), where memory consumption exceeds that of full fine-tuning. These findings clarify the limitations and trade-offs of FP8 for LoRA fine-tuning, and we will incorporate this analysis into the manuscript.
>
> ### **W5: In the background on LoRA: you may want to mention that the adapters can be downcast.**
>
> $\to$ Thank you for the suggestion. We will clarify that by adding “While prior methods focus on integrating LoRA to a quantized model, it is also possible to quantize the low rank adapters themselves, such as [C1].”
>
> [C1] Guo, Hang, et al. "Intlora: Integral low-rank adaptation of quantized diffusion models." arXiv 2410.21759 (2024).
>
> ### **W7: I would say what your method does earlier, at least at a high level.**
>
> $\to$ We will add high-level description after the introduction, such as “We leverage FP8 arithmetics to accelerate the LoRA finetuning task, where we maximize the speedup by not keeping a physical LoRA weight. Instead, we use the quantization error space and shape it to imitate the effect of a LoRA update. Then, the update is migrated and merged to the original weight using top-k row selection, saving both the memory and the expensive update cost.”
>
> ### **W8: Typo: Line 84, you say $\tilde W$ is high precision**
>
> $\to$ We will correct that as follows: $DQ(\cdot)$ reconstructs the high-precision approximation of quantized weight ($\tilde W$).
>
> ### **W10: Worth mentioning LoftQ and LQLoRA on initialization from quantization error.**
>
> $\to$ Thank you for the suggestion. LoftQ is cited after Eq. 9 (line 195), and we will additionally reference LQLoRA in the revised manuscript.
>
> ### **W11: Error bars for evaluation.**
>
> $\to$ Here are the experimental results with two-sigma error bars.
>
> **Table R2:** LLaMA-7B Experimental results with error bars
> |Method|Time per Step|Alpaca|2-sigma error|OASST1|2-sigma error|
> |---|---|---|---|---|---|
> |FP16|2.87s|0.3456|$\pm$0.0130|0.3656|$\pm$0.0164|
> |QLoRA|5.45s|0.3278|$\pm$0.0099|0.3524|$\pm$0.0125|
> |QA-LoRA|9.44s|0.3472|$\pm$0.0065|0.3627|$\pm$0.0036|
> |IR-QLoRA|8.27s|0.3488|$\pm$0.0031|0.3652|$\pm$0.0035|
> |TorchAO|2.18s|0.3401|$\pm$0.0132|0.3611|$\pm$0.0218|
> |FP6-LLM (E2M3)|2.72s|0.2351|$\pm$0.0073|0.2479|$\pm$0.0321|
> |FP6-LLM (E3M2)|2.72s|0.2301|$\pm$0.0018|0.2452|$\pm$0.0157|
> |Fishman et al.|2.29s|0.3445|$\pm$0.0095|0.3605|$\pm$0.0259|
> |FALQON|1.79s|0.3471|$\pm$0.0017|0.3469|$\pm$0.0051|
> ||
>
> Following the NeurIPS paper checklist, we provide two-sigma error bars of our experimental results. Table R2 presents the MMLU accuracy on LLaMA-7B architecture. The experimental results show that our method shows consistent performance. Due to the character constraints, we provide partial results with error bars. However, we will put more experimental results with error bars in the manuscript.
>
> ### **W12: Excessive 244MB uncompressed code and not all .git folders were deleted.**
>
> $\to$ The immense size is from the large contents in the 3rd-party libraries (Transformers, PEFT, lm-evaluation-harness), which we modified for our framework. Please consider the large size of the code as our efforts to provide runnable code to the reviewers. Also, we found that the .git folder of the lm-evaluation-harness library is not deleted. However, it is from the public and popular GitHub repository. Therefore, it does not harm the double-blind policy of the conference.
>
> ### **Q1: What is special about FP8? Compared to say, INT4/NF4?**
>
> $\to$ FP8 quantization accelerates **fine-tuning** by enabling faster matrix multiplications, leveraging native GPU support for both weights and activations. In contrast, INT4 and NF4 quantization are primarily used for reducing **inference** memory and are typically weight-only. They do not speed up fine-tuning since computations still use high-precision arithmetic. Thus, FP8 uniquely enables significant training acceleration, which is the focus of our work.
>
> ### **Q2: Figure 1: What is behind the dashed lines, theoretical maximum, and pre-training speed up? What is the empirical setup?**
>
> $\to$ The theoretical maximum of FP8 is the Tensor Core throughput ratio of GPU (e.g., 330.3TFLOPS/165.2TFLOPS = 2.0 for RTX4090), pretraining speed up is reported from LLaMA-70B pretraining with 8xH100 GPUs, and the empirical speed up is measured speed of LoRA layers with TorchAO baseline on an RTX4090 GPU.
>
> ### **Q3: The sign of the error term flips from equation 8 to 9, keep it as negation?**
>
> $\to$ We respectfully argue that equations 8 and 9 are mathematically correct; therefore, the sign should be flipped. The sign is flipped because we did transposition of the error term $\Delta_Q W$ at equation 9: $DQ(\tilde W) + \Delta_Q W = W $ (Eq. 8) $\Rightarrow DQ(\tilde W) = W - \Delta_Q W$. However, if we misunderstood the reviewer’s question, please feel free to add a comment and elaborate on this.
>
> ### **Q4: Line 184-185: I don’t see another interpretation as possible, what do you mean?**
>
> $\to$ We mentioned that the key idea of melded LoRA interprets quantization error $\Delta_Q W$ as an additive tensor to the original high-precision weight $W$. We agree that it is a well-known interpretation of quantization error, and we apologize for the possible confusion.
>
> ***
> We appreciate the reviewer for the constructive review. We believe our rebuttal addresses the reviewer’s concerns and better highlights the contributions of our paper. We would be happy to discuss if there are any additional concerns regarding our rebuttal.

---

> > ### Comment · Reviewer_EcqX · 2025-08-05
> >
> > Thanks for your answers, I don't havey any other questions but will think about the final score in the next day or so.

---

> > > ### Author Response · Authors · 2025-08-06
> > >
> > > Thank you for the response. We are glad that all the concerns have been addressed. If any further questions come up, we would be happy to discuss.

---

### Official Review · Reviewer_CKwV · 2025-07-02

**Clarity:** 2
**Significance:** 3
**Originality:** 3
**Rating:** 3
**Confidence:** 4

**Summary:**

This paper identifies an interesting performance bottleneck in reduced-precision large language model (LLM) fine-tuning: while low-bit formats like FP8 can significantly speed up large matrix multiplications, their inherent quantization overheads lead to a net slowdown when applied to the small-dimensional matrices used in Low-Rank Adaptation (LoRA). To address this, the authors propose FALQON, a framework that eliminates the need for a separate computational path for LoRA adapters. The core idea is to "meld" the LoRA adapter directly into the FP8-quantized backbone weights from the start, using the initial quantization error as a form of adapter initialization. The framework reformulates the forward and backward passes to compute gradients efficiently within this merged structure and introduces a selective, row-wise update mechanism to apply only the most significant changes. The experimental results demonstrate that FALQON can achieve up to a 3x training speedup over existing quantized LoRA methods while maintaining comparable accuracy on various benchmarks.

**Questions:**

- The row-wise update mechanism introduces a top-k operation on the gradient matrix ΔB at every training step. This operation is not free. Could you provide an analysis of the computational overhead of this top-k selection and discuss how it impacts the overall wall-clock time speedup claimed by FALQON?

- The dimensions (e.g., rank r) for the micro-benchmark in Figure 1a are not specified.

- The trend in Figure 1b, where efficiency does not improve with increasing rank, is counter-intuitive and contradicts the paper's own analysis without further explanation.

**Ethical Concerns:**

["NO or VERY MINOR ethics concerns only"]

**Final Justification:**

Thank you for the hard work in preparing the rebuttal. While I see potential in the proposed methods for reducing LoRA PEFT's memory requirements, the additional results (Table R3) reveal a concerning accuracy trade-off. Compared to FP16 and QA-LoRA, there's noticeable performance degradation, suggesting that FALCON's weight update approach needs refinement. I remain concerned whether the proposed methods can achieve fine-tuning performance comparable to the FP16 LoRA baseline.

I agree that the inefficiency stems from the LoRA algorithm itself. However, given the rebuttal's emphasis on system-level benefits (e.g., memory savings), I believe a systems-focused venue like MLSys or DAC would be more appropriate. Therefore, I maintain my original rating.

**Limitations:**

Yes

**Quality:**

3

**Strengths And Weaknesses:**

Strengths
- This paper pinpoints the less-explored inefficiency of FP8 acceleration for LoRA finetuning. It was interesting to note that LoRA would suffer from quantization overhead due to its small dimensional arithmetic.

- The central idea of "melded LoRA" seems to be interesting as it would avoid the separate LoRA path entirely.


Weaknesses
- The paper's central premise—the need to quantize the LoRA path to FP8—feels somewhat constructed. LoRA adapters are intentionally small, and standard FP16 LoRA is a very strong and widely used baseline. The motivation for forcing the entire pipeline into an FP8 format, which creates the very problem this paper solves, is not sufficiently justified. The paper would be stronger if it presented compelling use cases (e.g., specific hardware constraints, or scenarios where training thousands of adapters simultaneously makes the FP16 overhead non-trivial) where this end-to-end FP8 acceleration is a critical, practical need rather than an academic exercise.

- The most significant weakness is the omission of the standard FP16 LoRA as a performance baseline in the results tables (Tables 1, 2, 3). FALQON introduces several approximations (e.g., initializing from quantization error, fixing the adapter matrix A, and only updating B, avoid maintaining full-precision weight) that could potentially degrade fine-tuning quality. Without a direct comparison to the de-facto standard, it is impossible to assess whether the impressive speedup comes at a hidden cost to model performance. The claim of maintaining "comparable accuracy" is only validated against other quantized methods, which may already be degraded.

- The paper suffers from a lack of clarity in its experimental reporting, hindering a full understanding of the results.
  - The experimental settings for baselines like QLoRA and TorchAO are ambiguous. The precision of the backbone, the LoRA adapters (presumably FP16 for QLoRA?), and optimizer states should be explicitly stated for all methods.
  - The explanation for why TorchAO causes an OOM on the 13B model while FALQON does not is unclear. If FALQON's key contribution is to improve LoRA efficiency, how come it is more memory efficient than TorchAO (which does not have LoRA at all)?
- The work's contribution is heavily weighted towards systems-level optimization and implementation efficiency rather than fundamental algorithmic advances in machine learning. While it involves algorithmic changes, the core novelty lies in overcoming a hardware-specific bottleneck. As such, the paper might be a more natural fit for a systems-focused conference like MLSys or DAC, rather than a general ML conference like NeurIPS.

---

> ### Author Rebuttal · Authors · 2025-07-31
>
> We thank the reviewer for the constructive review. We are delighted that the reviewer acknowledged that the motivation and the central idea are interesting.
>
> Here, we address the reviewer’s concerns and questions. This is a brief summary of the rebuttal.
>
> * **W1-Use cases for FP8 LoRA**: We elaborate on real-world use cases (4 tasks from 6 papers) in the manuscript and further present cost comparison of FP16 and FP8 in rebuttal.
> * **W2-Accuracy comparison with FP16**: We show comparison with FP16 LoRA and FALQON also provides comparable accuracy.
> * **W3.1/Q2-Detailed experimental settings**: We provide more detailed experimental settings.
> * **W3.2-Why TorchAO causes OOM in 13B model?**: TorchAO store weights as FP16 which takes more memory than FALQON.
> * **Q1-Computational overhead regarding top-k selection**: We provide further analysis, and top-k selection incurs approximately 0.5% overhead of wall-clock time.
> * **Q3-Efficiency does not improve with increasing rank in the motivational study**: The efficiency starts to improve after 4k rank (which is infeasible).
>
> We will include this in the manuscript and thank the reviewer for the effort.
> The following is a detailed rebuttal.
>
> ***
> ### **W1: The motivation for FP8 format is not sufficiently justified. The paper would be stronger if it presented compelling use cases (e.g., scenarios where training thousands of adapters simultaneously makes the FP16 overhead non-trivial).**
>
> $\to$ We respectfully highlight that large-scale training of LoRA adapters is crucial to real-world applications such as personalization [54, 50, 26], multitask learning [44], domain adaptation [11], and multilingual summarization (see lines 111–118). For example, PLoRA [54] targets millions of users, and iLoRA [C1] evaluates up to 6 million users. In these settings, FP16 overhead becomes significant.
>
> To further support this, Tables R1 and R2 provide a direct cost comparison on the MovieLens-1M [C2] scenario with 6,040 users, showing that FALQON reduces training time by months and cost by thousands of dollars compared to FP16 LoRA. This demonstrates the real-world benefits of our approach.
>
> **TABLE R1:** Comparison of financial costs for fine-tuning LoRA
> |\$ USD|QLoRA|QA-LoRA|FALQON|\||vs QLoRA|vs QA-LoRA|
> |---|---:|---:|---:|---|---|---|
> |RTX4090|\$6,001|\$10,328|\$1,971|\|| $\Downarrow$ \$4,030  |$\Downarrow$ \$8,357 |
> |L40S|\$35,126|\$58,603|\$10,070|\||$\Downarrow$ \$25,057|$\Downarrow$ \$48,533|
> |H100|\$41,122|\$33,114|\$13,419|\||$\Downarrow$ \$27,703|$\Downarrow$ \$19,695|
> ||
>
> **TABLE R2:** Comparison of temporal costs for fine-tuning LoRA
> |Time (days, 8 GPU)|QLoRA|QA-LoRA|FALQON|\||vs QLoRA|vs QA-LoRA|
> |---|---:|---:|---:|---|---|---|
> |4090|89.3|153.7|35.7|\||$\Downarrow$ 53.6|$\Downarrow$ 118.0|
> |L40S|98.3|164.0|37.7|\||$\Downarrow$ 60.6|$\Downarrow$ 126.3|
> |H100|31.1|25.1|13.3|\||$\Downarrow$ 17.9|$\Downarrow$ 11.8|
> ||
>
> [C1] Kong, Xiaoyu, et al. "Customizing language models with instance-wise lora for sequential recommendation." NeurIPS (2024).
>
> [C2] Harper, F. Maxwell et al. "The movielens datasets: History and context." ACM TiiS (2015).
>
> ### **W2: The most significant weakness is the omission of the standard FP16 LoRA as a performance baseline in the results tables (Tables 1, 2, 3). The claim of maintaining "comparable accuracy" is only validated against other quantized methods.**
>
> $\to$ Thank you for this suggestion. We have included FP16 LoRA results below for direct comparison:
>
> **Table R3:** Comparison with FP16 LoRA
> |Model|Method|Time per Step|Alpaca (MMLU)|OASST1 (MMLU)|
> |---|---|---|---|---|
> |LLaMA-7B|FP16|2.87s|0.3527|0.3656|
> ||QLoRA|5.45s|0.3272|0.3564|
> ||QA-LoRA|9.44s|0.3548|0.3609|
> ||IR-QLoRA|8.27s|0.3388|0.3605|
> ||FALQON (Ours)|1.80s|0.3491|0.3481|
> ||
> |LLaMA-13B|FP16|OOM|0.4641|0.4714|
> ||QLoRA|9.37s|0.4443|0.4605|
> ||QA-LoRA|18.02s|0.4729|0.4769|
> ||IR-QLoRA|14.46s|0.4349|0.4620|
> ||FALQON (Ours)|3.26s|0.4644|0.4645|
> ||
>
> The results show FALQON has competitive accuracy, even compared to FP16 LoRA. Please note that FP16 LoRA could not run on a single RTX4090 for LLaMA-13B due to its large memory consumption. Therefore, we run FP16 LoRA on an RTX A6000 GPU and only report accuracy without time per step (noted as OOM). Our focus on quantized methods is motivated by resource constraints common in practical deployments, but we agree that including FP16 LoRA enhances the comparison, and we will add these results to the manuscript.
>
>
> ### **W3.1: The experimental settings for baselines like QLoRA and TorchAO are ambiguous.**
>
> $\to$ We followed the original settings for each baseline: QLoRA and IR-QLoRA use NF4 for the backbone and FP16 for adapters; QA-LoRA uses INT4 for the backbone and FP16 for adapters. For FP8-based methods (TorchAO, Fishman et al., and ours), both the backbone and adapters use FP8. All experiments use a 32-bit paged AdamW optimizer. We will clarify these details in the revised manuscript. Thank you for the suggestion.
>
> ### **W3.2: The explanation for why TorchAO causes an OOM on the 13B model while FALQON does not is unclear. How come FALQON is more memory efficient than TorchAO?**
>
> $\to$ TorchAO stores master weights in FP16, so it does not reduce memory compared to FP16 LoRA. While TorchAO performs dynamic quantization for computation, FALQON maintains all weights in FP8, enabling significant memory savings. Consequently, FALQON can fine-tune the 13B model on a single RTX4090 GPU, which is not feasible with TorchAO or FP16 LoRA.
>
> ### **W4: The work's contribution is heavily weighted towards systems-level optimization and implementation efficiency rather than fundamental algorithmic advances in machine learning. While it involves algorithmic changes, the core novelty lies in overcoming a hardware-specific bottleneck. As such, the paper might be a more natural fit for a systems-focused conference like MLSys or DAC, rather than a general ML conference like NeurIPS.**
>
> $\to$ We respectfully disagree. The core idea of this paper is an algorithmic advance of the traditional quantization method by innovating the LoRA fine-tuning procedure. In addition, our target problem (the quantization bottleneck) is not a hardware-specific bottleneck, but an algorithm-induced slowdown. Because the quantization algorithm delivers the bottleneck, we improve the quantized training framework by modifying the LoRA algorithm. Lastly, quantization is one of the well-discussed topics in general-domain ML conferences such as NeurIPS. We believe our paper is beneficial to the general ML audience of the NeurIPS conference by providing practical values of efficient fine-tuning algorithms.
>
> ### **Q1: Could you provide an analysis of the computational overhead of this top-k selection and discuss how it impacts the overall wall-clock time speedup claimed by FALQON?**
>
> $\to$ The top-k selection step accounts for only about 0.5% of total training time (see Table R4), as it operates over $r$ rows (with $r \ll d$).
>
> **Table R4:** Top-k selection overhead of LLaMA-7B and 13B models
>  |Time (ms)|Train Step|Top-k selection|Overhead|
> |---|:---:|:---:|:---:|
> |LLaMA-7B|1769.09|9.97|0.56%|
> |LLaMA-13B|3210.36|13.85|0.43%|
> ||
>
> Moreover, top-k selection reduces the update matrix dimension from $r$ to $k$, which lowers the weight update cost (See Figure 2 and Algorithm 3-line 6 in the paper.). As shown in Table R5, this leads to an overall speedup in step time:
>
> **Table R5:** Comparison of time per steps with and without top-k selection
> |Time per Step (ms)|w/ Top-k|w/o Top-k|Speed up|
> |---|:---:|:---:|:---:|
> |LLaMA-7B|1769.09|1814.98|1.026x|
> |LLaMA-13B|3210.36|3307.48|1.030x|
> ||
>
> Thus, top-k selection has negligible overhead and actually contributes to overall training speedup.
>
> ### **Q2: The dimensions (e.g., rank r) for the micro-benchmark in Figure 1a are not specified.**
>
> $\to$ Thank you for the comments. The rank is 64, which is a default setting of all our experiments, unless mentioned otherwise.
>
> ### **Q3: The trend in Figure 1b, where efficiency does not improve with increasing rank, is counter-intuitive and contradicts the paper's own analysis without further explanation.**
>
> $\to$ The efficiency does improve when we increase the rank over 1024 (Table R6).
>
> **Table R6:** Efficiency analysis across various LoRA ranks
> | |Rank:|16|32|64|128|256|512|1024|2048|4096|8192|16384|
> |---|---|---|---|---|---|---|---|---|---|---|---|---|
> |LLaMA-7B|Speed Up|0.50$\times$|0.55$\times$|0.44$\times$|0.48$\times$|0.43$\times$|0.62$\times$|0.69$\times$|0.60$\times$|0.80$\times$|**1.18$\times$**|**1.22$\times$**|
> |FT: 49.01GB|LoRA Mem.|0.67*GB*|0.78*GB*|0.98*GB*|1.39*GB*|2.20*GB*|3.83*GB*|7.09*GB*|13.60*GB*|26.63*GB*|**52.70*GB***|**104.82*GB***|
> |LLaMA-13B|Speed Up|0.55$\times$|0.59$\times$|0.50$\times$|0.52$\times$|0.46$\times$|0.48$\times$|0.64$\times$|0.61$\times$|0.88$\times$|**1.21$\times$**|**1.20$\times$**|
> |FT: 95.48*GB*|LoRA Mem.|1.31*GB*|1.51*GB*|1.91*GB*|2.70*GB*|4.29*GB*|7.46*GB*|13.81*GB*|26.50*GB*|51.89*GB*|**102.66*GB***|**204.21*GB***|
> ||
>
> The results show that the efficiency starts to increase after 1024 rank and becomes beneficial in ranks 8192 and 16384. However, the memory consumption on those ranks exceeds that of full fine-tuning, making LoRA inefficient at all.
>
> We believe that this is not counter-intuitive as the quantization overheads scale in $O(n^2)$ while the computation of matrix multiplication scales in $O(n^3)$. Therefore, in the small-dimensional region (widely used rank 16-512), the efficiency does not seem to increase due to its exponentially scaling characteristics. We will modify the motivational study to the extended range and add this explanation.
>
> ***
> We thank the reviewer for the valuable comments. We believe that this rebuttal effectively addresses the reviewer’s concerns and clarifies the paper's contributions. Also, we would be happy to address any additional comments the reviewer may provide.

---

> > ### Author Response · Authors · 2025-08-07
> >
> > We appreciate your thoughtful comments. We did our best to address concerns and questions from the review. If there are any remaining questions or points requiring clarification, we would be glad to discuss them. As the discussion period comes to a close, we remain available for any further communication at your convenience.

---

> > ### Comment · Reviewer_CKwV · 2025-08-08
> >
> > Thank you for the hard work in preparing the rebuttal. While I see potential in the proposed methods for reducing LoRA PEFT's memory requirements, the additional results (Table R3) reveal a concerning accuracy trade-off. Compared to FP16 and QA-LoRA, there's noticeable performance degradation, suggesting that FALCON's weight update approach needs refinement. I remain concerned whether the proposed methods can achieve fine-tuning performance comparable to the FP16 LoRA baseline.
> >
> > I agree that the inefficiency stems from the LoRA algorithm itself. However, given the rebuttal's emphasis on system-level benefits (e.g., memory savings), I believe a systems-focused venue like MLSys or DAC would be more appropriate. Therefore, I maintain my original rating.

---

> > > ### Author Response · Authors · 2025-08-08
> > >
> > > We thank you for the comments. We address the reviewer’s remaining concerns. Here is a brief summary.
> > >
> > > * **Concern 1-Accuracy compared to FP16**: The average drop is only **0.69\%p**, while even **higher accuracy** were reported in Table R3.
> > > * **Concern 2-Venue suitability**: Our work targets a core challenge for the general NeurIPS community. NeurIPS values practical contributions with system-level benefits, as shown by papers like QLoRA and the FlashAttention series.
> > >
> > > The following is a detailed clarification.
> > >
> > > ***
> > > ### **Concern 1: Noticeable accuracy drop compared to FP16**
> > >
> > > $\to$ We would like to respectfully argue that we have achieved comparable accuracy compared to FP16 LoRA. As shown in Table R3, the average accuracy degradation is only **0.69\%p**. Notably, FALQON even outperforms FP16 in a setting (LLaMA-13B, Alpaca dataset: FP16-0.4641, FALQON-0.4644). From the experimental results, we see FALQON show comparable accuracy, and we believe this small trade-off is acceptable given the significant speed and memory gains.
> > >
> > >
> > > ### **Concern 2. Venue suitability.**
> > >
> > > $\to$ We respect the reviewer’s opinion that FALQON could fit a system-focused venue. However, we believe NeurIPS is the most appropriate forum for FALQON. **Our method addresses a core challenge for the broader ML community**: enabling memory- and compute-efficient fine-tuning of large models on consumer-level hardware, which is especially relevant for researchers and practitioners with limited resources (such as academia, small research groups, etc.).
> > >
> > > Importantly, **the original purpose of LoRA [19] is to provide system-level benefits** such as memory savings and faster fine-tuning. Following this, baselines targeting memory savings—LoRA (ICLR 2022), QLoRA (NeurIPS 2023 Oral), QA-LoRA (ICLR 2024), and IR-QLoRA (ICML 2024 Oral)—were all published at top ML conferences to benefit general audiences. Thus, system-level benefits make FALQON suitable for NeurIPS, directly supporting the need for scalable and accessible LoRA PEFT solutions.
> > >
> > > Additionally, NeurIPS has a strong tradition of publishing works that advance practical machine learning, such as the FlashAttention series [C3-C5] (NeurIPS 2022/2024 Spotlight and ICLR 2024). These works demonstrate that NeurIPS values contributions offering memory savings and faster training or inference, highlighting the community’s interest in efficient and accessible solutions with system-level benefits.
> > >
> > >
> > > [C3] Dao, Tri, et al. Flashattention: Fast and memory-efficient exact attention with io-awareness. NeurIPS 2022
> > >
> > > [C4] Dao, Tri. FlashAttention-2: Faster Attention with Better Parallelism and Work Partitioning. ICLR 2024
> > >
> > > [C5] Shah, Jay, et al. Flashattention-3: Fast and accurate attention with asynchrony and low-precision. NeurIPS 2024
> > >
> > > ***
> > > We hope these clarifications address your concerns and demonstrate the relevance and value of our contributions to the NeurIPS community.

---

### Official Review · Reviewer_kFb2 · 2025-07-06

**Clarity:** 3
**Significance:** 4
**Originality:** 3
**Rating:** 5
**Confidence:** 5

**Summary:**

The paper analyzes the benefits and overhead of FP8 utilization for low-rank quantization. The authors observed that the speed-up gained by using FP8 is offset by the quantization overhead due to the small dimension of low-rank adapters. To address this issue, the paper proposes the FALQON framework, which interleaves an FP8-quantized backbone with LoRA adapter models. This integration requires modifications to the forward path and backward pass for gradient computation, as proposed in the paper. The results show an improvement of 1.8x to 3x for LLM models compared to using LoRA and FP8 quantization separately. An ablation study over various training parameters such as learning rate, batch size, and number of ranks is also presented.

**Questions:**

In the Efficient Gradient Computation section, it is mentioned that gradients are computed only for matrix B, which is empirically efficient for effective fine-tuning. Could you elaborate on this point in more detail and explain why gradient computation for matrix A is not required?

**Ethical Concerns:**

["NO or VERY MINOR ethics concerns only"]

**Final Justification:**

In the rebuttal, the authors evaluate the trade-off between performance and the number of quantized LoRA adapters, demonstrating the consistent speed-up performance of FALQON compared to QLoRA. Moreover, they provide FP4 fine-tuning experiments, showing improvements over FP8 LoRA quantization. Furthermore, my question regarding the bypassing of the calculation of gradients for matrix A was thoughtfully addressed. Overall, the approaches presented in this paper are both valuable and novel.

**Limitations:**

Yes

**Paper Formatting Concerns:**

There are no concerns regarding the paper format.

**Quality:**

4

**Strengths And Weaknesses:**

Strengths:

1- The proofs for the claims in the paper are presented clearly.

2- The idea behind the efficient gradient calculation by combining the gradients of the weights and the low-rank matrix (A) is novel and significant.

Weaknesses:

1- The new approach seems to be valuable when multiple agents use multiple quantized LoRA adapters, as mentioned in the paper. It is suggested that multiple LoRA adapter fine-tunings are evaluated on a specific benchmark in the paper, demonstrating the trade-off between performance and the number of quantized LoRA adapters.

2- FP4 fine-tuning has been suggested in previous papers [1,2,3], and it would be helpful to include it in the experiments section as a low-bit FP quantization baseline for comparison with the proposed approach.

[1] Liu, Lian, et al. "COMET: Towards Practical W4A4KV4 LLMs Serving." Proceedings of the 30th ACM International Conference on Architectural Support for Programming Languages and Operating Systems, Volume 2. 2025.

[2] Wang, Jie, et al. "Fp4-quantization: Lossless 4bit quantization for large language models." 2024 IEEE International Conference on Joint Cloud Computing (JCC). IEEE, 2024.

[3] Roy, Somnath. "Understanding the impact of post-training quantization on large language models." arXiv preprint arXiv:2309.05210 (2023).

---

> ### Author Rebuttal · Authors · 2025-07-31
>
> We appreciate valuable feedback from the reviewer. We are encouraged that the reviewer finds our work novel and significant, with well-presented content.
>
> We address the reviewer’s concerns and incorporate suggestions. Here is a brief summary of this rebuttal.
>
> * **W1-Suggesting evaluation of trade-off between performance and the number of LoRA adapters**: We provide experimental results that present robust performance gains.
> * **W2-Extension to FP4 fine-tuning**: We provide estimated speed up of FP4, where FALQON show 2.15$\times$ faster fine-tuning than FP16.
> * **Q1-Request for the rationale of freezing A matrix**: Updating A matrix has small-to-no effects on the model quality [1], but incurs significant overhead.
>
> We will incorporate these into the manuscript and thank the reviewer for the time.
> The following is a detailed rebuttal.
>
> ***
>
> ### **W1: The new approach seems to be valuable when multiple agents use multiple quantized LoRA adapters. It is suggested that multiple LoRA adapter fine-tunings are evaluated on a specific benchmark in the paper, demonstrating the trade-off between performance and the number of quantized LoRA adapters.**
>
> $\to$ Thank you for the comments. We believe the reviewer is asking for scalability on training multiple LoRA adapters at the same time.
> To address this, we provide an analysis of the trade-off between performance and the number of LoRA adapters in Table R1.
>
> **Table R1:** Trade-off between performance and the number of LoRA adapters.
> |# of LoRA adapters:|1|2|4|8|16|32|64|
> |---|---|---|---|---|---|---|---|
> |QLoRA|3.99*s*|4.00*s*|4.00*s*|4.04*s*|4.19*s*|4.48*s*|6.82*s*|
> |FALQON|1.76*s*|1.77*s*|1.77*s*|1.77*s*|1.80*s*|2.34*s*|2.93*s*|
> |Speed Up|2.27$\times$|2.26$\times$|2.26$\times$|2.28$\times$|2.33$\times$|1.91$\times$|2.33$\times$|
> ||
>
> This experiment directly evaluates the scenario mentioned by the reviewer, demonstrating that FALQON consistently outperforms the baseline in terms of time per step, even as the number of LoRA adapters increases.
>
> However, when we say “training numerous adapters” in our paper, we refer to running multiple independent fine-tuning jobs, each producing a single adapter. As far as we know, there is no use case to train multiple LoRA weights attached to the same backbone, simultaneously. Thus, we also provide experiments on training multiple LoRA weights independently. Table R2 reports the total training time required for such large-scale personalization, using the MovieLens-1M [C2] dataset scenario from [C1], which involves 6,040 users.
>
> **TABLE R2:** Comparison of temporal costs for fine-tuning LoRA
> |Time (days, 8 GPU)|QLoRA|QA-LoRA|FALQON|\||vs QLoRA|vs QA-LoRA|
> |---|---:|---:|---:|---|---|---|
> |4090|89.3|153.7|35.7|\||$\Downarrow$ 53.6|$\Downarrow$ 118.0|
> |L40S|98.3|164.0|37.7|\||$\Downarrow$ 60.6|$\Downarrow$ 126.3|
> |H100|31.1|25.1|13.3|\||$\Downarrow$ 17.9|$\Downarrow$ 11.8|
> ||
>
> These results confirm that FALQON provides significant temporal savings for both concurrent and large-scale fine-tuning scenarios.
>
> We hope these results address the reviewer’s concerns. If further clarification or discussion is needed, we welcome additional questions in the discussion phase.
>
> [C1] Kong, Xiaoyu, et al. "Customizing language models with instance-wise lora for sequential recommendation." NeurIPS (2024).
>
> [C2] Harper, F. Maxwell, and Joseph A. Konstan. "The movielens datasets: History and context." ACM TiiS (2015).
>
>
> ### **W2: It would be helpful to add FP4 fine-tuning baselines.**
>
> $\to$ Thank you for this helpful suggestion. To address it, we provide a comparison in Table R3 below. The FP4 speedups are estimated by applying the FP4/FP8 matrix multiplication throughput ratio, as specified in NVIDIA GPU documentation and available hardware benchmarks:
>
> **Table R3:** Speed up Comparison for FP8 and FP4 compared to FP16
> |Model|Method|FP8|FP4|
> |---|---|---|---|
> |LLaMA-7B|TorchAO|1.31$\times$|1.55$\times$|
> ||FALQON|1.61$\times$|2.15$\times$|
> ||
> |LLaMA-13B|TorchAO|1.21$\times$|1.35$\times$|
> ||FALQON|1.51$\times$|1.91$\times$|
> ||
>
> These results suggest FP4 could provide further speedup over FP8 for both TorchAO and FALQON, assuming similar implementation efficiency. However, current hardware and software support for full FP4 fine-tuning is still limited. We are actively seeking access to FP4-capable resources and will report empirical results as they become available.
>
> Additionally, we would like to clarify that prior works [1,2,3] primarily focus on weight-only FP4 quantization for **inference** rather than end-to-end fine-tuning. Most of them do not speed up fine-tuning since computations still use high-precision arithmetic or do not support backward or update of weights. Recent papers such as [C3] and [C4] do address FP4 fine-tuning, but they target full-rank LLM training and focus mainly on large-dimensional matrix multiplications. In these scenarios, quantization overheads remain significant, and we believe FALQON’s approach would continue to offer benefits even in the FP4 regime.
>
> [C3] Castro, Roberto L., et al. “Quartet: Native FP4 Training Can Be Optimal for Large Language Models.” arXiv preprint arXiv:2505.14669 (2025).
>
> [C4] Chmiel, Brian, et al. “FP4 All the Way: Fully Quantized Training of LLMs.” arXiv preprint arXiv:2505.19115 (2025).
>
> ### **Q1: Could you explain why gradient computation for matrix A is not required?**
>
> $\to$ It is because updating matrix A does not significantly contribute to model quality [49], while slowing down the training. The earlier work, LoRA-FA [49], empirically showed that freezing matrix A has little effect on the final quality of fine-tuning.
>
> In addition to LoRA-FA, we conduct further experiments showing the effect of freezing matrix A in Table R4. The results empirically verify that we may freeze matrix A during fine-tuning without sacrificing much of the model quality. Also, computing the gradient of matrix A affects the training speed, showing approximately a 16% slowdown (13B). This is because gradient of matrix $A$ ($\frac{\partial \mathcal{L}}{\partial A}$) is computed as $B^{\top}\frac{\partial \mathcal{L}}{\partial O} x^{\top}$, which cannot be pre-calculated unlike matrix $B$. We thank the reviewer for their insightful comments, and we will clarify this in the revised version of the manuscript.
>
> **Table R4:** Comparison of training speed and model quality regarding matrix A update
> |Model|Method|Time per Step|Alpaca (MMLU)|OASST1 (MMLU)|
> |---|---|---|---|---|
> |LLaMA-7B|FALQON|1.80s|0.3491|0.3481|
> |LLaMA-7B|+ Update matrix A|1.94s|0.3469|0.3470|
> ||
> |LLaMA-13B|FALQON|3.26s|0.4644|0.4645|
> |LLaMA-13B|+ Update matrix A|3.89s|0.4634|0.4656|
> ||
>
> [49] Zhang, Longteng, et al. "LoRA-FA: Memory-efficient low-rank adaptation for large language models fine-tuning." arXiv preprint arXiv:2308.03303 (2023).
>
> ***
> We thank the reviewer for their insightful comments. We believe this rebuttal addresses the reviewer’s concerns and further clarifies our contribution. We welcome further questions or comments and would be happy to discuss them.

---

> > ### Comment · Reviewer_kFb2 · 2025-08-05
> >
> > I appreciate the authors' response to my comments, which is complete and thoughtful. My questions and comments have been addressed, and I have raised my score to 5.

---

> > > ### Author Response · Authors · 2025-08-06
> > >
> > > We again appreciate the thoughtful review and are glad our response addressed the concerns. Thank you for the positive update and for acknowledging our responses. We will incorporate the suggestions into the revised version of our paper.

---

### Comment · Area_Chair_6axN · 2025-08-06
**Reminder: Participate in Author Discussions Before Submitting “Mandatory Acknowledgement”**

Dear Reviewers,

Thank you for your contributions during the reviewer–author discussion period. If you have not yet engaged with the authors, please do so at your earliest convenience.

As per NeurIPS 2025 policy, reviewers must participate in discussions with authors before submitting the “Mandatory Acknowledgement.” If you have already submitted the acknowledgement without engaging in discussion, we kindly request that you participate in the discussion now.

Your input at this stage is essential to ensuring a fair and thorough review process.
Thank you again for your dedication to the review process.

Best regards,
AC

---

### Note · Authors · 2025-08-12

We appreciate all reviewers and the AC for the careful review and discussion, and we thank the AC for encouraging author–reviewer discussion. During the discussion, kFb2, EcqX, and UHn5 indicated their concerns were addressed and provided positive responses.
***
### **Summary of Contributions**
This paper accelerates FP8-quantized LoRA fine-tuning on consumer-level hardware. We show that FP8’s quantization overheads yield slowdowns for small-dimensional matrices, limiting LoRA speedup. FALQON removes LoRA’s quantization overhead, delivering approximately **3$\times$ faster training** with comparable accuracy.
***
### **Strengths from the Review**
* ***Novel and interesting idea*** [kFb2, CKwV, EcqX]
* ***Clear presentation and experiments*** [kFb2, EcqX, UHn5]
* ***Identifies under-explored topic of FP8 overhead on small matrices*** [CKwV]
* ***Improves computational efficiency, which is important and strengthens the significance*** [EcqX, UHn5]
***
### **Addressed Concerns**
* ***Practical scenario and use cases*** [kFb2, CKwV, UHn5]: We present practical scenarios where FALQON saves up to $48,533 and 126 days.
* ***Rank-acceleration trade-off*** [CKwV, EcqX, UHn5]: Baseline helps LoRA only at extremely high ranks ($\ge$ 8192), which is infeasible.
* ***FP8 vs. INT4/NF4 baselines*** [EcqX, UHn5]: FALQON accelerates fine-tuning of LoRA, while baselines target inference speed.
* We also provide ***FP4 extension** [kFb2], **matrix A update results** [kFb2], **key differences to baselines** [CKwV], **top-k overhead analysis** [CKwV], **error bars** [EcqX], **loss curves** [UHn5], and **responses to writing suggestions***.
***
### **Further Discussion Topics**
We would like to mention the remaining concerns (See our final comments for CKwV).
* ***Accuracy compared to FP16***: We achieved comparable accuracy to FP16. The average degradation is only 0.69%p, where *FALQON even outperforms FP16* in a setting.
* ***Venue suitability***: FALQON addresses a crucial ML challenge and benefits the NeurIPS community and resource-limited researchers. LoRA’s original goal was also system-level gains (i.e., memory and speed), and ML conferences valued quantized LoRA baselines. Also, NeurIPS’s record of recognizing practical advances (e.g., FlashAttention) makes it an ideal forum for FALQON.
***
We believe these final remarks summarize our contributions and clarifications. We will incorporate these into the paper. We again thank the reviewers and AC for their efforts.

---

### Decision · Program_Chairs · 2025-09-17

**Decision:**

Accept (poster)

**Comment:**

This paper proposes FALQON, a framework to accelerate LoRA fine-tuning by using FP8 arithmetic. The key idea is to merge LoRA adapters into FP8-quantized backbone weights, avoiding the overhead that makes FP8 inefficient for small matrices and introducing a tailored gradient method for the merged design. The paper accomplishes this by viewing quantization error as an additive component of quantized weights, which allows LoRA adapters to be included in the FP8-quantized weights. Because the adapters are merged into the backbone, standard autograd does not directly compute their gradients. The authors address this by introducing a customized gradient computation strategy tailored to the melded structure, which further reduces overhead. After extensive rebuttals and new experiments introduced by the authors, reviewers appreciated the problem focus, experiments, and improvements over prior art.

Comments:

1. Expand the caption for Fig 1 to cater to a more general AI/ML audience. For instance, this caption can mention finetuning explicitly and detail how these speed-up trendlines were evaluated. Also, "Up-Proj" and "Down-Proj" are not defined in the paper.
2. Include all new experiments that were introduced in the rebuttals to improve the manuscript.